# Cryo-EM structure of human O-GlcNAcylation enzyme pair OGT-OGA complex

Ping Lu ⓘ[1,2,3,5], Yusong Liu ⓘ[2,3,4,5], Maozhou He ⓘ[2,3], Ting Cao[2,3], Mengquan Yang[1,2,3], Shutao Qi[2,3], Hongtao Yu ⓘ[2,3]✉ & Haishan Gao ⓘ[2,3]✉

O-GlcNAcylation is a conserved post-translational modification that attaches N-acetyl glucosamine (GlcNAc) to myriad cellular proteins. In response to nutritional and hormonal signals, O-GlcNAcylation regulates diverse cellular processes by modulating the stability, structure, and function of target proteins. Dysregulation of O-GlcNAcylation has been implicated in the pathogenesis of cancer, diabetes, and neurodegeneration. A single pair of enzymes, the O-GlcNAc transferase (OGT) and O-GlcNAcase (OGA), catalyzes the addition and removal of O-GlcNAc on over 3,000 proteins in the human proteome. However, how OGT selects its native substrates and maintains the homeostatic control of O-GlcNAcylation of so many substrates against OGA is not fully understood. Here, we present the cryo-electron microscopy (cryo-EM) structures of human OGT and the OGT-OGA complex. Our studies reveal that OGT forms a functionally important scissor-shaped dimer. Within the OGT-OGA complex structure, a long flexible OGA segment occupies the extended substrate-binding groove of OGT and positions a serine for O-GlcNAcylation, thus preventing OGT from modifying other substrates. Conversely, OGT disrupts the functional dimerization of OGA and occludes its active site, resulting in the blocking of access by other substrates. This mutual inhibition between OGT and OGA may limit the futile O-GlcNAcylation cycles and help to maintain O-GlcNAc homeostasis.

O-GlcNAcylation entails the covalent attachment of O-GlcNAc to the hydroxyl group of serines or threonines of target proteins[1–4]. It modifies thousands of human proteins involved in diverse cellular processes and localized to major cellular compartments, including the nucleus, the cytoplasm, and mitochondria[5,6]. Protein O-GlcNAcylation is a dynamic and reversible process, which is mediated by a single pair of enzymes, OGT and OGA (Fig. 1a, b)[1,2]. OGT contains an N-terminal tetratricopeptide repeat (TPR) domain required for substrate recognition and a C-terminal glycosyltransferase family B domain (GTD) that catalyzes the modification of protein substrates with uridine

diphosphate GlcNAc (UDP-GlcNAc) as the sugar donor[7–9]. OGA consists of an N-terminal glycoside hydrolase domain (GHD) that removes O-GlcNAc from substrates, a stalk domain, and a C-terminal histone acetyltransferase-like domain[7,10–12]. OGA forms a functional homodimer in which the catalytic domain from one monomer is capped by the stalk domain of the other to generate the substrate-binding cleft[7,10–13].

UDP-GlcNAc is the product of the hexosamine biosynthetic pathway that links the metabolism of glucose, amino acids, fatty acids, and nucleotides[1,2,4,14]. Its production is sensitive to changes in

[1]College of Life Sciences, Zhejiang University, Hangzhou, Zhejiang, China. [2]New Cornerstone Science Laboratory, School of Life Sciences, Westlake University, Hangzhou, Zhejiang, China. [3]Westlake Laboratory of Life Sciences and Biomedicine, Hangzhou, Zhejiang, China. [4]School of Life Sciences, Fudan University, Shanghai, China. [5]These authors contributed equally: Ping Lu, Yusong Liu. ✉e-mail: yuhongtao@westlake.edu.cn; gaohaishan@westlake.edu.cn

glycolysis, amino acid synthesis, fatty acid levels, and nucleotide levels[1,14–16]. Thus, protein O-GlcNAcylation serves as an important nutrient sensor[1,14,16]. In response to nutrient availability and various stress conditions, O-GlcNAcylation modulates the structure, stability, localization, and function of target proteins, thereby regulating the cell cycle, signal transduction, gene transcription, and protein translation, among other cellular processes[2,14,16]. Well-characterized

O-GlcNAcylation substrates include the tumor suppressor TP53[17], TGFβ activated kinase 1 binding protein 1 (TAB1)[18], histones[19], the histone methyltransferase EZH2[20], the microtubule-binding protein TAU[21], neuronal protein α-synuclein[22], and the m6A mRNA-binding protein YTHDF1[23]. Aberrant O-GlcNAcylation has been implicated in human metabolic syndromes and chronic diseases, including cancer, diabetes, cardiovascular diseases, and neurodegenerative diseases[24–27].

**Fig. 1 | Cryo-EM structures of human OGT and OGT–OGA complex. a** The dynamic protein O-GlcNAcylation cycle mediated by OGT and OGA. **b** Domains and motifs of human OGT and OGA. TPR, tetratricopeptide repeat. **c** Human HEK293 cells with the endogenous OGT locus tagged with the HaloTag were treated with the HaloPROTAC3 compound for different duration times. Total cell lysates were blotted with the indicated antibodies. The experiment was repeated thrice with similar results. **d** Cryo-EM map of human OGT dimer (left panel) and the structure of the OGT dimer fitted into the EM map (right panel). GTD, glycosyltransferase family B domain. **e** Close-up views of the OGT dimerization interface.

**f** O-GlcNAcylation of recombinant TAB1 by OGT WT or its monomeric 4A mutant in the presence of UDP-GlcNAc. The relative O-GlcNAc levels were quantified and indicated below. The experiment was repeated thrice with similar results. **g** Cryo-EM map of human OGT-OGA complex (left panel) and the structure of the OGT-OGA complex fitted into the EM map. GHD, glycoside hydrolase domain; IDR, intrinsically disordered region. **h** Superimposition of the structures of the OGT-OGA complex and the OGT dimer (colored in gray and with only one protomer shown). **i** Cryo-EM density of the UDP and NAG in OGT-OGA complex. Source data are provided as a Source Data file.

In addition, missense mutations of the OGT TPR domain cause X-linked intellectual disability (XLID), which is characterized by poor adaptive behavior and severe cognitive disability to understand complex ideas[28–30].

Biochemical and structural studies on human OGT and OGA have revealed the catalytic mechanisms of their enzymatic activities towards peptide substrates[8,10–13,31]. However, how OGT selects its native (but not peptide) substrate and maintains substrate selectivity remains a long-standing question in the field. In addition, it is still unknown how OGT and OGA coordinate their enzymatic activities towards diverse substrates to achieve optimal O-GlcNAcylation levels in response to the nutrient status[2,6,7]. Paradoxically, human OGA has been reported to directly interact with its opposing enzyme OGT[32–36]. The nature and functional importance of this interaction remain unexplored, however.

Here, we reconstitute the full-length human OGT-OGA complex and determine its cryo-EM structure. Structural and biochemical studies not only provided the molecular basis for how OGT selects its substrate OGA, but also revealed that OGT and OGA enzymatically inhibit each other in vitro. Neither OGT nor OGA in the OGT-OGA complex can act on other substrates. The direct mutual inhibition between this enzyme pair at the protein level might limit futile O-GlcNAc cycling and contribute to the maintenance of steady-state levels of O-GlcNAcylation.

## Results

### Cryo-EM structure of human OGT dimer
Using CRISPR-Cas9 genome editing technology, we knocked in the Halo-Flag Tag into the endogenous OGT locus in HEK293 cells (Supplementary Fig. 1a, b). The addition of the HaloPROTAC compound induced the efficient and rapid degradation of the resulting OGT-Halo-Flag fusion protein[37] (Fig. 1c). Expectedly, the overall cellular O-GlcNAc levels were significantly reduced upon OGT depletion, confirming the essential role of OGT in cellular O-GlcNAcylation[2,38,39].

We expressed and purified full-length human OGT (Supplementary Fig. 1c). Recombinant wild-type OGT, but not its catalytically deficient K852M mutant[8,40], catalyzed efficient O-GlcNAcylation of the full-length human TAB1 in a UDP-GlcNAc-dependent manner, indicating that the recombinant OGT protein was active (Supplementary Fig. 1d). We next determined the structure of OGT at an overall resolution of 3.69 Å using single-particle cryo-EM (Fig. 1d, Supplementary Fig. 2, and Supplementary Fig. 3). As previously reported[41], OGT formed a scissor-shaped dimer (Fig. 1d). The TPR domain of each monomer consists of 27 antiparallel α-helices, which stack together in a right-handed superhelical conformation with a lumen of 22 Å in diameter (Fig. 1d). The conserved asparagine residues line along the lumen and align in a ladder-like configuration to engage substrates. Structural comparison with previously solved OGT structures reveals that the catalytic domain (GTD$_{477–1046}$) in all structures adopts an almost identical conformation (Supplementary Fig. 3c). Consistent with earlier studies[9,41], TPR6 and TPR7 of OGT mediate its dimerization (Fig. 1e). Mutations of four conserved residues at the dimer interface to alanine (OGT 4A, W208A/L209A/I211A/H212A) disrupted OGT dimerization, based on size exclusion chromatography and cryo-EM analysis (Supplementary Fig. 4a, b). Similar to the previous reports that showed the dimerization-defective OGT exhibited lower O-GlcNAc transferase

activity towards substrates[9,41], here OGT 4A mutant was less active in catalyzing O-GlcNAcylation of both TAB1 and YTHDF1 here in vitro (Fig. 1f and Supplementary Fig. 4c, d), indicating that OGT dimerization promotes optimal modification of certain substrates[41].

### Cryo-EM structure of the OGT-OGA complex
OGT and OGA have been shown to directly interact with each other[33,36,42]. In addition, depletion of OGA in human cells could also co-deplete OGT[43]. As a substrate of OGT, due to the limited structural information, how OGA is recognized and modified by OGT remains underexplored. In vitro, recombinant human OGT and OGA co-fractionated on size exclusion chromatography (Supplementary Fig. 5a), indicating that they could form a complex. We then determined the cryo-EM structure of the OGT-OGA complex with an overall resolution of 3.92 Å (Fig. 1g, Supplementary Fig. 5b–d, and Supplementary Fig. 6).

OGT in the OGT–OGA complex has a scissor-like dimeric architecture similar to that of OGT alone (Fig. 1g and Supplementary Movie 1). In the complex, the distance between the two active sites in the dimer (located at the handles of the scissor) is shorter compared to OGT alone: from ~90 Å in OGT alone to ~80 Å in the OGT-OGA complex (Fig. 1d, g). OGT TPRs 1–5 become more rigid and compacted upon OGA binding in the complex (Fig. 1h). In contrast to the apo OGT dimer that has no bound nucleotides, the OGT GTD domain in the OGT–OGA complex has a bound UDP in its active site (Fig. 1i).

In the cryo-EM map, only the N-terminal GHD of OGA and two segments of the intrinsically disordered region (IDR) following this domain were observed and could be modeled in the complex (Fig. 1g). The stalk and HAT-like domains of OGA were absent in the cryo-EM maps, presumably due to conformational flexibility. Crystal structures of the N-terminal fragment of OGA have revealed that the catalytic GHD forms a functional homodimer[10–13]. Interestingly, the OGA GHD dimerization interface is involved in OGT binding, and consequently the domain-swapped OGA dimer is disrupted in the OGT-OGA complex. The monomeric OGA docks on the convex surface of TPR11–13 in one OGT protomer (Fig. 2a, b). Residues 377–393 of the OGA IDR form a helix, which packs against the GHD. A long segment of the OGA IDR (residues 394–442) occupies the active site of the catalytic GTD and the entire lumen of the TPR domain of OGT (Fig. 2b). While no density was observed for a second OGA GHD at the equivalent site on the other OGT monomer, there was an obvious density belonging to the OGA IDR in the lumen of the TPR domain of this OGT molecule (Fig. 1g, Supplementary Fig. 5d and Supplementary Movie 2), suggesting that both OGT molecules are bound by OGA.

### OGA recognition by human OGT
OGA can interact with OGT in vivo and is a known native substrate of OGT[32–36]. Our co-immunoprecipitation (co-IP) assays confirmed that OGT and OGA pulled down each other when co-expressed in human HEK293 cells (Supplementary Fig. 9a, b). In the structure of OGT-OGA complex, the OGA fragment (377–442) containing a long intrinsic disordered region (IDR) dominates the direct association with OGT (Fig. 2b). Residues 394–407 of OGA interact with the GTD of OGT, while residues 408–442 of OGA mainly contact the TPR domain of OGT (Fig. 2b). The binding of OGA to OGT was confirmed by GST pull-down

assays using recombinant proteins (Fig. 2c). S405 of OGA lies in the proximity of UDP-GlcNAc bound at the OGT active site (Fig. 1i and Fig. 2d). The amide group of S405 forms a hydrogen bond with the α-phosphate of UDP, a critical interaction required for the transfer of GlcNAc to the residue to be modified[7]. The GlcNAc moiety is surrounded by active site residues H508, A664, K852, and H930 (Fig. 1i, Fig. 2d and Supplementary Fig. 6c), as observed in crystal structures of OGT bound to peptide substrates[8,31,40,44]. The strong electron density connecting the hydroxyl group of S405 and GlcNAc suggests that OGA S405 is glycosylated (Fig. 1i, Fig. 2d, and Supplementary Fig. 6c). Indeed, the catalytic deficient mutant OGA D175N, but not its S405A mutant, was modified by OGT in vitro, indicating that OGA S405 can be O-GlcNAcylated by OGT (Fig. 3f). Thus, our cryo-EM structure likely

captures the post-catalytic OGT-OGA complex, in which the GlcNAc moiety is attached to the hydroxyl group of S405.

Residues 408–442 of OGA adopt an extended conformation, and occupy the entire substrate-binding lumen of the TPR domain in OGT, through an array of non-covalent interactions, including hydrogen bonds and extensive non-bonded interactions (Fig. 2a, b and Fig. 3a–c). Notably, the backbone amide or carbonyl groups of OGA form hydrogen bonds or favorable polar interactions with asparagine residues in the TPR domain of OGT, including N94 and N97 in TPR3, N128 in TPR4, N165 in TPR5, N196 in TPR6, N230 in TPR7, N264 in TPR8, N298 in TPR9, N332 in TPR10, N366 in TPR11, N403 in TPR12, and N434 in TPR13 (Fig. 2a, b, Fig. 3a–c, and Supplementary Fig. 7a–c). Most of these residues, except N97, N165, and N403, are conserved residues

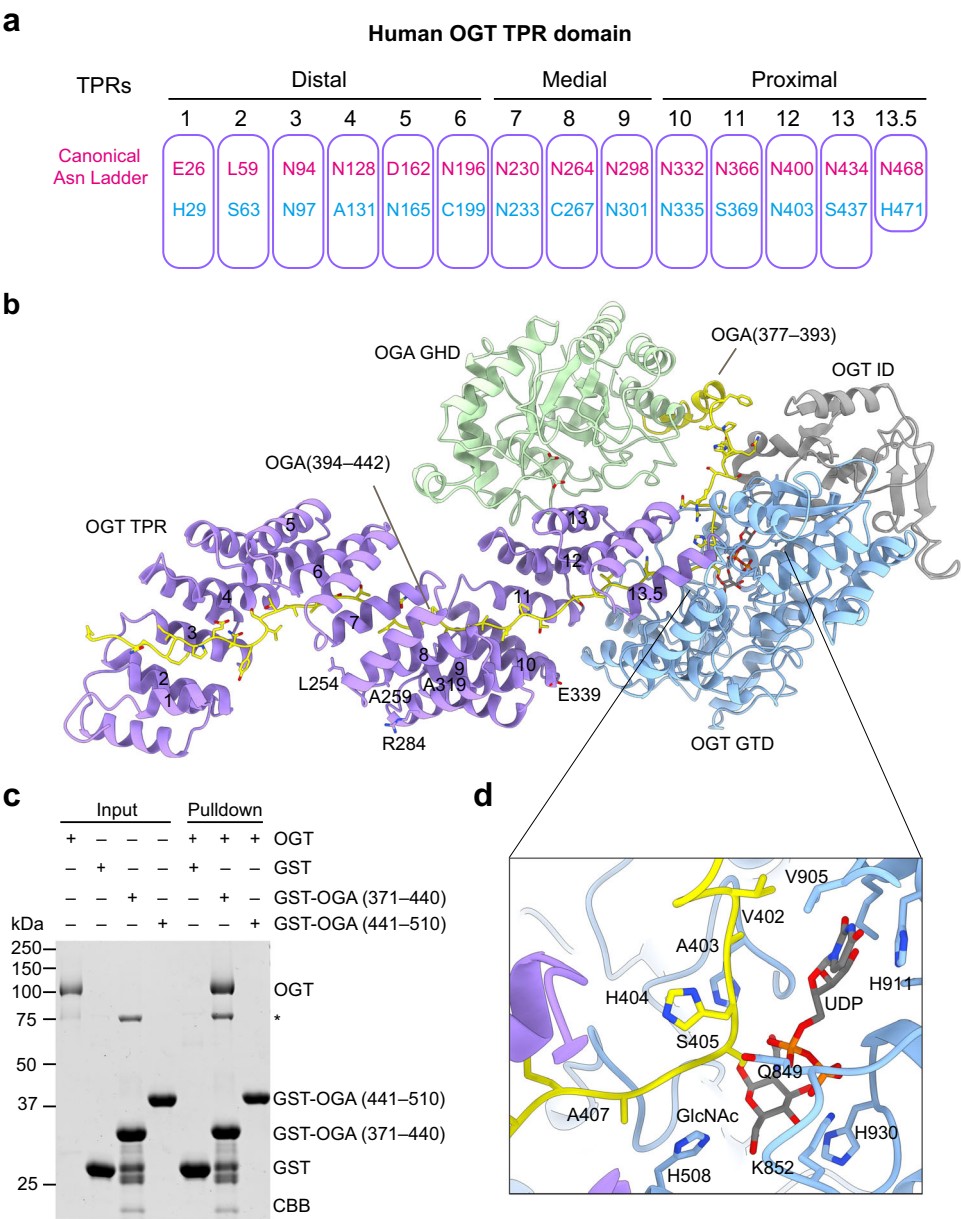

**Fig. 2 | OGA binding by human OGT. a** The schematic of human OGT TPR domain with the 13.5 tetratricopeptide repeats colored in purple. The distal, medial, and proximal units are indicated as shown. The residues at position 6 (top row, conserved Asparagine residues) and 9 (bottom row) in each unit are highlighted in red and blue, respectively. The canonical Asn Ladders are usually located at the inner concave of the TPR domain. **b** Extensive interactions between human OGT and OGA. XLID-related residues L254, A259, R284, A319, and E339 are shown as sticks

and labeled in the structure of the OGT-OGA complex. **c** Binding between full-length OGT and the indicated GST-OGA fragments. The input and proteins bound to beads were analyzed by SDS-PAGE followed by Coomassie blue staining. An asterisk indicates a contaminating protein. The experiment was repeated thrice with similar results. **d** Close-up view of OGT catalytic pocket (colored in purple) interacting with residues 402–407 of OGA (colored in yellow). Source data are provided as a Source Data file.

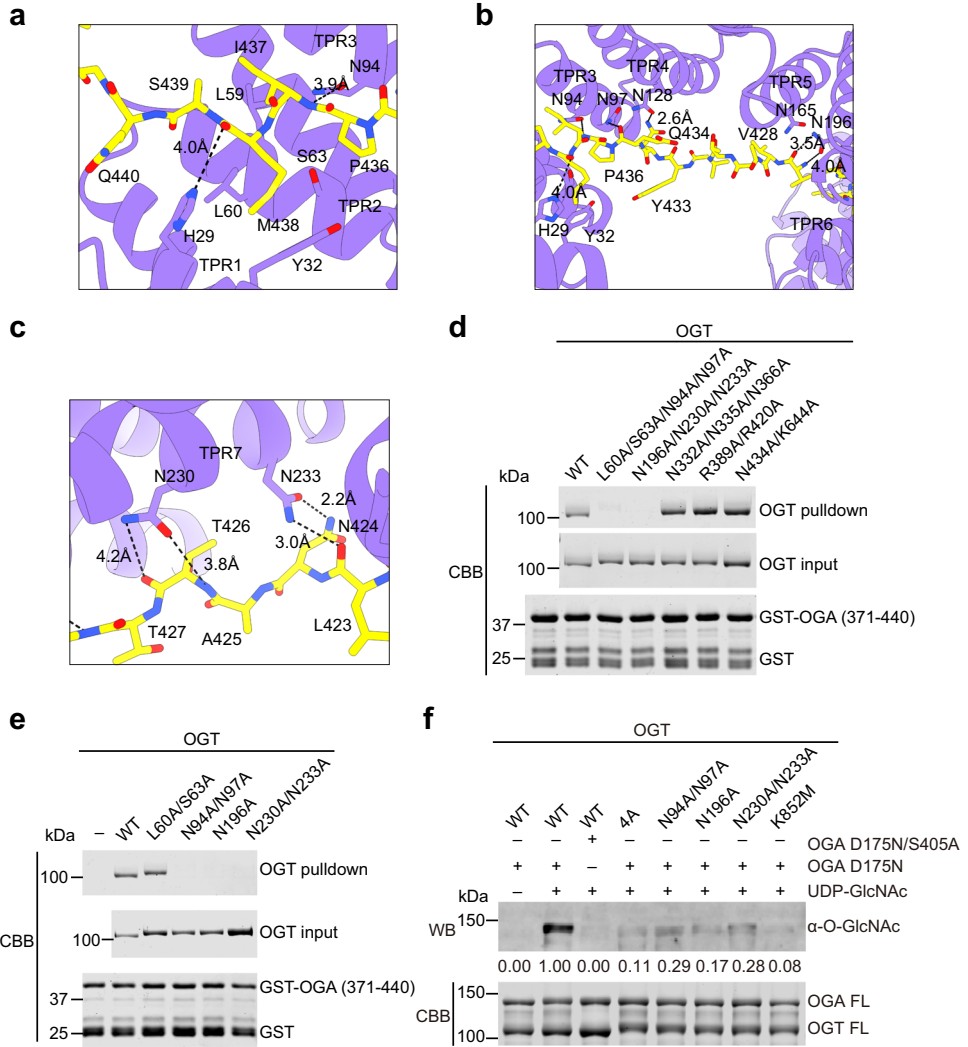

**Fig. 3 | Molecular details in OGA recognition by OGT TPR domain. a** Close-up view of OGT TPR 1–3 units (colored in purple) interacting with OGA (colored in yellow). **b** Close-up view of OGT TPR 3-6 units interacting with OGA. **c** Close-up view of OGT TPR7 interacting with OGA. **d** Binding between GST-OGA (residues 371–440) and the indicated OGT mutant proteins. The input proteins and proteins bound to GST beads were analyzed by SDS-PAGE and Coomassie staining. The experiment was repeated thrice with similar results. **e** Binding between GST-OGA

(residues 371–440) and OGT WT and the indicated TPR mutants. The experiment was repeated thrice with similar results. **f** O-GlcNAcylation assay of the catalytically inactive OGA mutant (D175N or D175N/S405A) by OGT wild type (WT) and the indicated mutants in the presence or absence of UDP-GlcNAc. The relative O-GlcNAc levels were quantified and indicated below. The experiment was repeated thrice with similar results. Source data are provided as a Source Data file.

from the asparagine ladder facing the TPR lumen[9,45] (Fig. 2a). Our findings are consistent with the crystal structures of OGT-HCF-1 PRO-repeat and OGT-TAB1 peptide, which showed that the conserved asparagine residues of OGT form a series of interactions with the amides of alternating residues along the peptide backbone of HCF-1/TAB1[40,46]. This mode of sequence-independent backbone recognition by OGT ensures that it can interact with a wide range of substrates, provided that these substrates have a long IDR like OGA. Some residues from the proximal TPRs, such as N400 and N403 in TPR12 and N468 in TPR13.5, interact with the side chains of D413 and S410 of OGA through hydrogen bonds (Supplementary Fig. 7c). These and other OGT-OGA interactions involving specific side chains of OGA confer the substrate specificity of OGT, consistent with previous studies which showed the active site of OGT imposes constraints on substrate sequence[40,44].

Mutations of certain OGT asparagine ladder residues at the observed OGT-OGA interfaces, such as N94 and N97 in TPR3, N196 in TPR6, and N230-N233 in TPR7, disrupted the interactions between OGT and OGA as revealed by co-IP assays from HEK293 cells and GST

pull-down assays in vitro (Fig. 3d, e and Supplementary Fig. 9a, b). Our findings are in overall agreement with a recent report using a GlcNAc electrophilic probe to study the recognition mode of OGA by OGT[36]. The OGA-binding-deficient OGT mutants were defective in catalyzing the O-GlcNAcylation of OGA in vitro (Fig. 3f). When overexpressed in HEK293 cells depleted of the endogenous OGT, the OGA-binding-deficient OGT mutants were less efficient in supporting the overall cellular O-GlcNAc levels (Supplementary Fig. 9c, d). These results not only validated the functional importance of the OGT-OGA interactions observed in our structure, but also suggested that the substrate recognition mode revealed by the structure of the OGT-OGA complex might be applicable to other OGT substrates in human cells. Previous studies have reported that an OGT mutant 5N5A (N332A/N366A/N400A/N434A/N468A) with mutations of the proximal asparagine ladder showed reduced glycosylation activity towards protein substrates in human cells[39,45,47]. Our results further established critical roles of the distal and medial asparagine ladders (including TPR3, TPR6, and TPR7) in substrate binding and modification.

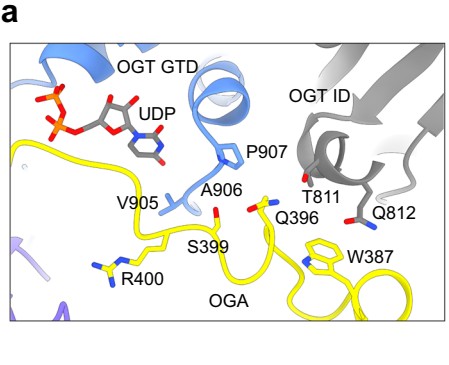

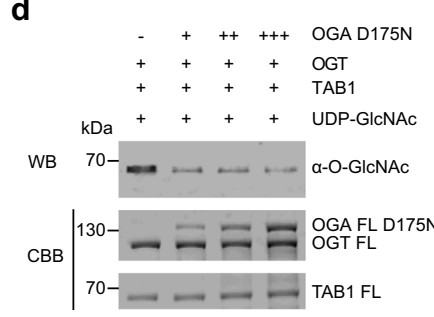

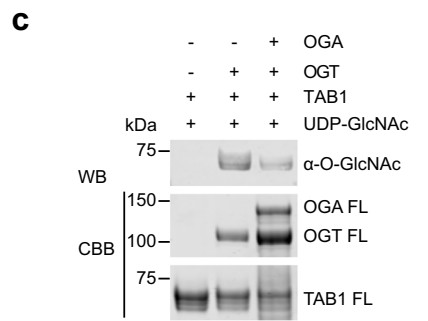

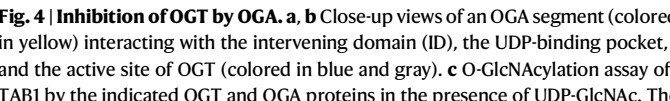

**Fig. 4 | Inhibition of OGT by OGA. a, b** Close-up views of an OGA segment (colored in yellow) interacting with the intervening domain (ID), the UDP-binding pocket, and the active site of OGT (colored in blue and gray). **c** O-GlcNAcylation assay of TAB1 by the indicated OGT and OGA proteins in the presence of UDP-GlcNAc. The experiment was repeated thrice with similar results. **d** O-GlcNAcylation assay of TAB1 added with different doses of OGA as a competitor. The relative O-GlcNAc levels were quantified and indicated below the gels in (**c, d**). The experiment was repeated thrice with similar results. Source data are provided as a Source Data file.

Mutations of some asparagine ladder residues, such as N332 in TPR10, N366 in TPR11, and N434 in TPR13, had no effect on OGA binding (Fig. 3d). Thus, not all asparagine ladder residues of OGT contribute to OGA binding equally. Binding hotspots in TPR3, TPR6, and TPR7 are more critical for interaction interactions between OGT and OGA (Fig. 3a–e), pointing to the key role of distal TPR units in OGA recognition. Mutations of N94 and N97 in TPR3 reduced the O-GlcNAcylation of TAB1 (Supplementary Fig. 8a). Mutation of N196 in TPR6 had a minor effect and mutations of N230 and N233 in TPR7 had virtually no effect. In contrast, mutations of N94 and N97 in TPR3 had little effect on the O-GlcNAcylation of YTHDF1, whereas mutations of asparagines in TPR6 and TPR7 impaired its O-GlcNAcylation (Supplementary Fig. 8b). These results indicate that OGT uses different binding hotspots for different substrates[45,47].

Mutations of the OGT TPR region, such as L254F, A259T, R284P, A319T, and E339G, are linked to XLID[29,48]. Most of these residues mutated in XLID are located at the convex side of the TPR lumen and do not directly contact OGA (Fig. 2b). These mutations are thus unlikely to affect OGA binding directly. They may instead affect the structural integrity of the OGT TPR domain and indirectly affect substrate binding and modification[29,48].

**Competitive inhibition of human OGT by OGA**
Enzyme-substrate interactions are typically transient, and modified substrates (i.e., products) are released to enable additional rounds of catalysis. Despite being modified by OGT, OGA remains bound to OGT after O-GlcNAcylation (Supplementary Fig. 5a), likely due to the slow dissociation rate between OGT and OGA. A segment of the OGA IDR occupies the entire substrate-binding lumen of the TPR domain of OGT and engages the asparagine ladder residues (Figs. 1g and 2b). Furthermore, OGA residues surrounding O-GlcNAcylated S405 maintain hydrophobic and hydrogen bonding interactions with the GTD and the intervening domain (ID) of OGT (Fig. 4a, b). Specifically, OGA W387 and Q396 directly contact T811 and Q812 in OGT ID, while Q396, S399, and R400 of OGA also engage the UDP-binding loop of OGT

(residues 905–907) (Fig. 4a). V402, A403, and H404 from OGA contact the UDP moiety (Fig. 4b). Thus, by occupying both the substrate-binding groove of the TPR domain and the active site of OGT (Fig. 2b, d, and Fig. 3a–c), OGA is expected to prevent OGT from binding and modifying other substrates.

Compared to OGT alone, the OGT–OGA complex was less efficient in catalyzing the O-GlcNAcylation of TAB1 (Fig. 4c). Importantly, addition of the OGA D175N mutant decreased O-GlcNAcylation of TAB1 by OGT (Fig. 4d). These results suggest that OGA can act as a competitive inhibitor of OGT in vitro.

**Structural basis for the inhibition of OGA by OGT**
Human OGA forms a domain-swapped homodimer, in which the C-terminal helix (residues 676–694) from one monomer docks on the stalk domain of the other monomer (Fig. 5a)[10,11,13]. In each monomer, the GHD packs against its own stalk domain, with the intervening IDR being absent from the structure (Fig. 5a)[10–12]. The glycopeptide substrate binds at a pocket formed by the GHD of one monomer and the stalk domain of the other monomer[11,12]. Thus, the dimerization of OGA is required for substrate recognition and subsequent removal of GlcNAc[10–12].

OGT binding triggers a dramatic conformational change of OGA (Fig. 1g and Supplementary Fig. 5d). Instead of the domain-swapped dimer, OGA in the OGT–OGA complex is monomeric (Figs. 1g and 2b). Only its GHD and a segment of the IDR is visible in the complex. The GHD docks onto the convex surface of TPR11 and TPR12 of OGT whereas the IDR binds to the lumen of the TPR domain of OGT (Fig. 2b). Superimposing the structures of the OGA dimer and the OGT–OGA complex reveals that OGT in the complex develops serious steric clashes with the stalk domain in the same OGA monomer and with the other OGA monomer (Fig. 5b, c). Thus, OGT binding is incompatible with OGA dimerization. Indeed, OGA V255, Y286, and D287, which are required for OGA dimerization, are hijacked by V411 and Q412 from OGT TPR12 in the OGT-OGA complex (Fig. 5d). In addition, TPR11 and TPR12 of OGT completely occlude the catalytic residues D174 and D175

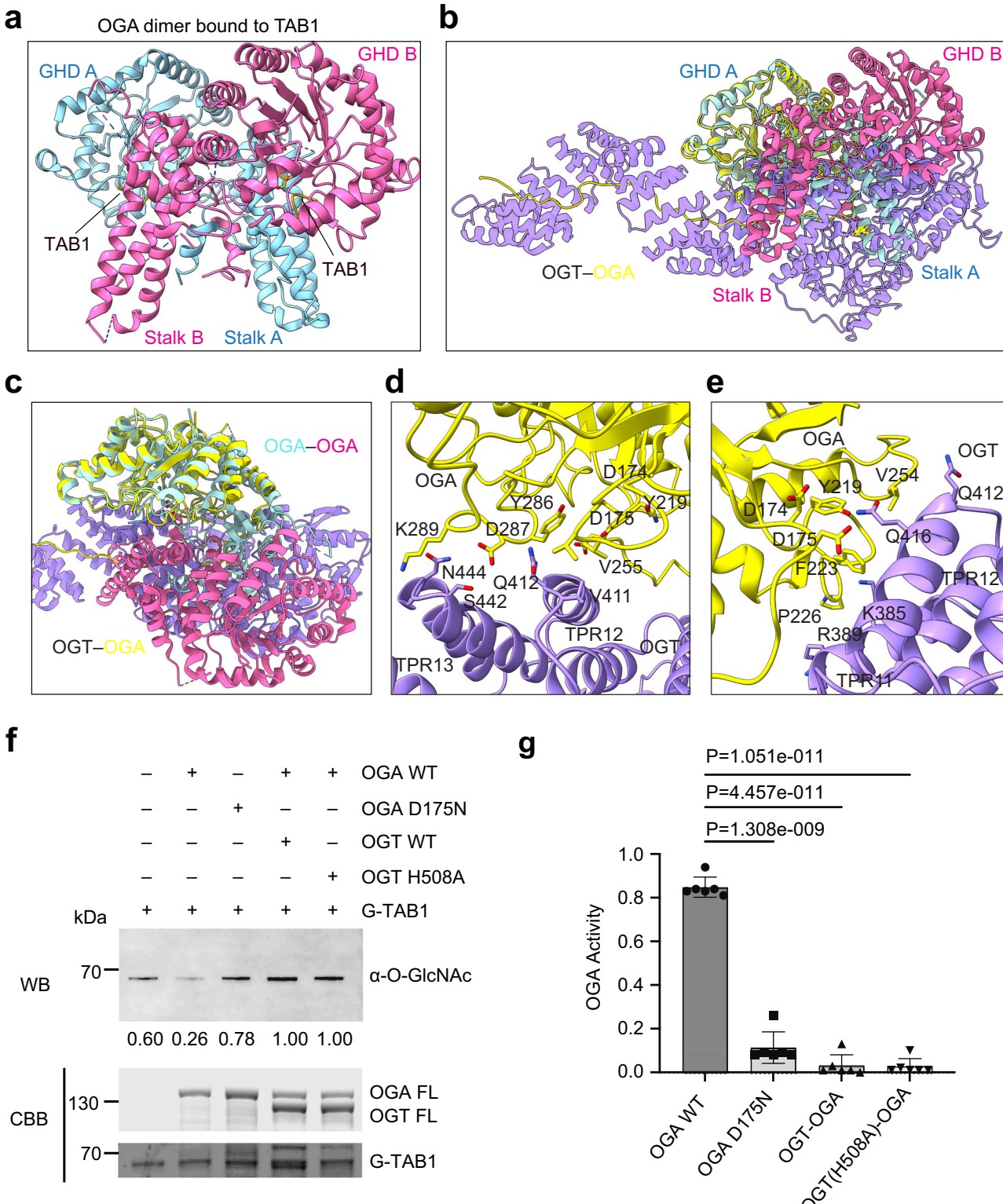

**Fig. 5 | Inhibition of OGA by OGT. a** Cartoon drawing of the crystal structure of the human OGA dimer (colored in cyan and pink) bound to a TAB1 peptide (PDB code: 5VVU). **b** Superimposition of the structures of the OGA molecules in the OGA dimer (as in **a**) and the OGT-OGA complex, showing the steric clashes between OGT and the second OGA molecule in the OGA dimer. **c** Close-up view of the superimposed structures in (**b**). **d, e** Close-up views of the interactions between OGT (colored in purple) and the GHD of OGA (colored in yellow). **f** Removal of TAB1 O-GlcNAcylation by OGA and its inhibition by OGT. O-GlcNAcylated TAB1 (G-TAB1) was incubated with OGA or the catalytically inactive OGA D175N mutant in the absence or presence of the indicated OGT proteins. The reaction mixtures were analyzed by SDS-PAGE, stained with Coomassie brilliant blue, and blotted with the anti-O-GlcNAc antibody. The relative O-GlcNAc levels were quantified and indicated below. The experiment was repeated thrice with similar results. **g** The quantitative OGA enzymatic assay using the artificial substrate PNP-GlcNAc with indicated $p$ values. Data were presented as mean ± SD, $n = 6$ biologically independent samples per condition. P value was calculated from an unpaired two-tailed $t$ test. Source data are provided as a Source Data file.

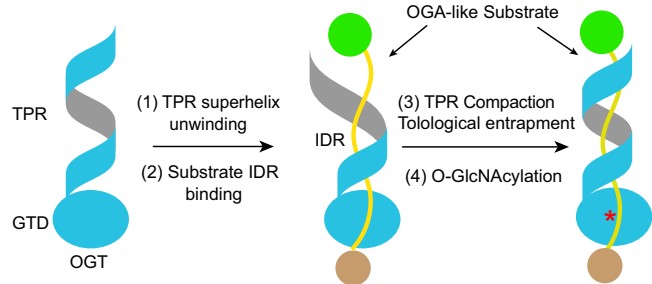

**Fig. 6 | Working model of substrate recognition and modification by human OGT.** For clarity, only one OGT monomer (colored in blue) is shown. OGT binds to the intrinsically disordered regions (IDRs) (colored in yellow) of OGA-like substrates through its TPR domain. The superhelical turns of the TPR domain have to partially unwind to allow the IDR to access its lumen. The superhelical turns then reform and wrap around the IDR in a topological embrace, which lengthens the lifetime of the OGT-substrate complex for optimal O-GlcNAcylation. TPR tetra-tricopeptide repeat, GTD glycosyltransferase domain, IDR intrinsically disordered region.

of OGA (Fig. 5e). The side chains of K385 and R389 from TPR11 contact the flexible stretch of T223 to P226 of OGA (Fig. 5e), which is very close to the substrate-binding cleft and the catalytic D loop (residues 172–182) of OGA[10–12]. OGT Q416 from TPR12 prevents OGA D175 from contacting Y219 (Fig. 5e), which stabilizes the activated conformation of D175 and mediates the rapid acid-and-base switch during the process of glycoside hydrolysis[10–12]. Therefore, the binding of OGT to OGA disrupts OGA dimerization and shields the active site and substrate-binding cleft of OGA.

OGA efficiently removed O-GlcNAcylation of TAB1 and YTHDF1, as evidenced by the weaker signals on the anti-O-GlcNAc blot of glycosylated TAB1 and YTHDF1 (Fig. 5f and Supplementary Fig. 10). Compared to OGA WT alone, the OGT–OGA complex had much weaker O-GlcNAcase activity towards either TAB1 or YTHDF1. In a quantitative OGA enzymatic assay, the OGT-OGA complex also had weaker activity against the artificial substrate p-nitrophenyl-beta-N-acetyl-glucosaminide (PNP-GlcNAc)[49] (Fig. 5g). OGA bound to the OGT H508A mutant also exhibited greatly decreased O-GlcNAcase activity (Fig. 5f, g and Supplementary Fig. 10), suggesting that O-GlcNAcylation of OGA by OGT is not required for OGA inhibition. Taken together, these data indicate that human OGT indeed inhibits the enzymatic activity of OGA in vitro.

## Discussion

The single pair of enzymes OGT and OGA regulates the dynamic cycling of O-GlcNAcylation on thousands of cellular proteins with diverse functions[14,16]. Our structural and functional analyses of the OGT-OGA complex provide key insights into how OGT recognizes its substrates. The GTD and ID of OGT interact with 9–10 residues N-terminal to the Serine/Threonine residue to be O-GlcNAcylated. The entire TPR domain of OGT, consisting of distal, medial, and proximal units, wraps around an extended 40-residue segment C-terminal to the modification site. Thus, remarkably, OGT can simultaneously engage up to 50 residues in an extended conformation in certain substrates. This interaction is reminiscent of the interactions between karyopherins and nuclear localization signals (NLS)[50], although the NLS segments are much shorter. Many asparagines in the asparagine ladder of OGT interact with the peptide backbone in a sequence-independent manner[40,44,46]. Different subsets of TPRs are required for the recognition of different substrates[7,45,47], suggesting that sparse binding hotspots involving sidechain interactions determine the substrate specificity of OGT. Although not all substrates engage the entire TPR domain of OGT, this type of backbone-dominated binding mode may enable the recognition of a wide variety of substrates by OGT.

Although the recognition modes and TPR hotspot requirement for different substrates may vary[7,41,45,47], OGT can act on a long segment of intrinsically disordered regions (IDRs) in substrates[45], as demonstrated in the case of OGA. Because the OGT TPR domain forms an α-solenoid with two complete superhelical turns, it needs to undergo substantial conformational changes for substrates to gain access to the lumen of all the TPRs, as seen in the OGT–OGA complex (Supplementary Movie 1 and Supplementary Movie 2). We propose that the TPR domain of OGT can transiently adopt a more extended conformation in which the superhelical turns of the α-solenoid are partially open and relaxed (Fig. 6). For OGA-like substrates that have IDRs, the IDR segment would first bind to a subset of TPRs. Other TPRs then form right-handed twists to wrap around a much larger IDR segment of the substrate. Once the substrate occupies the entire lumen of the TPR domain, OGT can then scan for optimal sequences within the IDR through passive diffusion and locate specific serines/threonines in this region for modification. If the IDR is located in between folded domains, the substrate cannot escape from the TPR lumen of OGT without the partial uncoiling of the superhelical α-solenoid. As such, the topological enclosure of substrates by the OGT TPR domain prolongs the lifetime of the enzyme-substrate complex and enables proper search-and-modification of target serines/threonines in one binding-release cycle.

The fact that the TPR domain of OGT becomes more rigid and compact upon OGA binding is generally in agreement with this model (Supplementary Movie 1 and Supplementary Movie 2). The most relaxed conformations of OGT are unlikely to be captured by our cryo-EM analyses. We hypothesize that the dimerization of OGT stabilizes the mid-sections of the two α-solenoids in the dimer, allowing each solenoid to transiently reach its more straightened conformation for substrate recruitment and release[41]. This might explain why OGT dimerization is required for the optimal O-GlcNAcylation of OGA-like substrates. Future biophysical experiments and molecular dynamics simulations are needed to test this hypothesis.

Being an important nutrient-sensing mechanism, O-GlcNAcylation integrates signals from several metabolic pathways[2,4]. Nutrient conditions that elevate UDP-GlcNAc levels generally increase global O-GlcNAcylation[14,16,51,52]. Under a specific nutrient condition, the functions of OGT and OGA need to be coordinated to maintain O-GlcNAc homeostasis. Previous studies have reported that the mutual regulation between OGT and OGA occurs at transcriptional levels[51–53]. Our discovery that OGT and OGA inhibit each other's enzymatic activity in vitro suggests a direct mechanism of mutual inhibition at the protein level.

Although OGT and OGA do not always form a stable complex and have their own individual interactomes in the cells[54], human OGA is well recognized as a substrate of OGT in vitro and in vivo[32–36], suggesting that in certain situations OGT molecules are bound to OGA, and vice versa. When bound to OGT, OGA can act as a gatekeeper and reduce the accidental O-GlcNAcylation of sub-optimal substrates. Conversely, OGA in the complex is inhibited by OGT and cannot remove O-GlcNAcylation of other substrates. We propose that the mutual inhibition between OGT and OGA may help to ensure the fidelity of O-GlcNAcylation, limit futile O-GlcNAc cycling, and maintain O-GlcNAc homeostasis. Future work on the regulation of the abundance and subcellular localization of the OGT-OGA complex by different nutrient states is needed to further explore the functional significance of this mutual inhibition and advance our understanding of this fascinating enzyme pair.

## Methods
### Protein expression and purification
The cDNAs encoding the full-length of human OGT (UniProt: O15294), OGA (UniProt: O60502), and YTHDF1 (UniProt: Q9BYJ9) were cloned from a human cDNA library. The cDNA encoding the full-length of human TAB1 (UniProt: Q15750) was synthesized by the Tsingke Biotech

Company. Human OGT was cloned into the pET21b vector (Novagen) to produce human OGT with a C-terminal His$_6$ affinity tag. Human OGA and TAB1 were individually cloned into the pRSF vector to produce proteins with an N-terminal His$_6$ and Trx tag followed by the PreScission protease site. The fragments of OGA (residues 371–440 and 441–510) were cloned into the pGEX-6P-1 vector. YTHDF1 was cloned into the pET28a vector with an N-terminal EGFP and a C-terminal His$_6$ affinity tag. Site-directed mutagenesis of human OGT and OGA was performed using the Q5® Site-Directed Mutagenesis Kit (New England Biolabs). OGT and its mutants were also cloned into the pCS2 expression vector with N-terminal Myc tags, and OGA was cloned into the pCS2 expression vector with N-terminal Flag tags for expression in human cells. All constructs were verified by DNA sequencing.

BL21 (DE3) Escherichia coli cells containing the desired plasmids were grown in LB medium with ampicillin (100 µg/ml for the pET21b and pGEX-6p-1 plasmids) or kanamycin (50 µg/ml for the pRSF plasmid) with shaking at 37 °C to OD$_{600}$ of 1.0. The cell cultures were cooled to 18 °C, induced with 0.5 mM IPTG for 12–15 h, and harvested by centrifugation. The cell pellets were suspended in the lysis buffer (25 mM Tris-HCl pH 8.0, 150 mM NaCl, 5 mM β-mercaptoethanol, 1 mM PMSF, and with 5 mM imidazole added for His$_6$-tagged proteins). The cells were lysed by sonication or an UltraHigh Pressure Homogenizer and centrifuged at 40,000 × $g$ for 50 min at 4 °C. The supernatants were incubated with pre-equilibrated Ni$^{2+}$-NTA agarose beads for His$_6$-tagged proteins (Qiagen) or Glutathione Sepharose 4B beads for GST-tagged proteins (GE Healthcare) for 2 h at 4 °C. The beads were washed with 20 column volumes (CV) of the wash buffer (25 mM Tris pH 8.0, 150 mM NaCl, 5 mM β-mercaptoethanol, with 20 mM imidazole added for His$_6$-tagged proteins). The His$_6$-tagged proteins were eluted with 15 ml elution buffer (25 mM Tris-HCl pH 8.0, 150 mM NaCl, 5 mM β-mercaptoethanol, and 250 mM imidazole). The GST-tagged proteins were eluted with 5 ml elution buffer (25 mM Tris-HCl pH 8.0, 150 mM NaCl, 5 mM β-mercaptoethanol, and 15 mM reduced GSH) or incubated with the HRV 3C protease overnight. The proteins were loaded onto a Resource Q column and fractionated by the AKTA Pure system (GE Healthcare) equilibrated with Buffer A (25 mM Tris pH 8.0, 10 mM NaCl, 5 mM β-mercaptoethanol). The proteins were eluted with a linear 5–50% gradient of Buffer B (25 mM Tris pH 8.0, 1 M NaCl, 5 mM β-mercaptoethanol) over 13 CVs. The pooled peak fractions were concentrated and loaded onto a Superose 6 10/300 Increasing Column equilibrated with Buffer C (25 mM HEPES pH 7.4, 100 mM NaCl, 2 mM DTT, 0.05% NP40). The peak fractions were collected, analyzed by 10% SDS-PAGE, aliquoted, and stored at -80 °C.

For producing the human OGT-OGA complex, OGT and OGA were co-expressed in BL21 (DE3) and purified using the same protocol as described above. For producing O-GlcNAcylated TAB1 (G-TAB1) or O-GlcNAcylated YTHDF1 (G-YTHDF1), TAB1/YTHDF1 and OGT were co-expressed in BL21 (DE3). G-TAB1 and G-YTHDF1 were purified using the same protocol as described above. TAB1/YTHDF1 O-GlcNAcylation was verified by Western blotting using the anti-O-GlcNAc antibody (RL2) (Abcam, ab2739). The peak fractions containing G-TAB1/YTHDF1 were collected and stored.

### Pull-down assays
For GST pull-down assays, the concentrations of all proteins used in the assays were adjusted to 1 mg/ml. GST or GST-OGA proteins were used as baits. His$_6$-tagged OGT and its mutant proteins were used as prey. Each 10 µl bait sample was incubated with 10 µl Glutathione Sepharose 4B beads for 1 h at 4 °C. The beads were washed 3 times with the binding buffer (25 mM Tris pH 8.0, 150 mM NaCl, 2 mM DTT, 0.05% NP40). Then 10 µl prey proteins were added and incubated with beads in a 100-µl binding reaction for 2 h at 4 °C. The beads were washed 5 times with the binding buffer and boiled with the SDS sample buffer. The samples were analyzed by SDS-PAGE, stained with Coomassie blue, and imaged using the LI-COR Odyssey system.

For in vivo Myc/Flag pull-down (Co-Immunoprecipitation) assay, 293FT cells were co-transfected with pCS2-Myc/pCS2-Myc-OGT WT/mutant plasmids with pCS2-Flag/pCS2-Flag-OGA using Lipofectamine 2000 (Thermo Fisher Scientific) and cultured for 48 h in a 37 °C incubator before collection by centrifuge. After cell lysis by sonication in TENT buffer (25 mM Tris pH 7.4, 150 mM NaCl, 0.5% TritonX100, 1 mM EDTA) with the addition of 1 mM PMSF, protease and phosphatase Inhibitor, the cell lysate was centrifuged, and supernatants were separately incubated with anti-Myc and anti-Flag magnetic beads (LABLEAD) for 2 h. After the removal of the unbound cell lysate by centrifuge, the beads were washed with TENT buffer and boiled with sample loading buffer for SDS-PAGE and western blot. The bait and bound prey were blotted with corresponding antibodies.

### O-GlcNAcylation and O-GlcNAcase assays
For the O-GlcNAcylation assays, OGT WT and mutants or the OGT–OGA complex and its mutants were incubated with human TAB1 for 90 min at 37 °C or human OGA proteins (WT, D175N, or D175N/S405A) overnight at 37 °C. The incubation time for the YTHDF1 protein was 30 min. The assays were performed in 20 µl volumes with 2 µg enzymes, 2 mM UDP-GlcNAc, 1 µg TAB1 or OGA or YTHDF1 in the reaction buffer (25 mM Tris pH 8.0, 100 mM NaCl, 5 mM MgCl$_2$, 1 mM DTT). The reactions were stopped by the addition of the SDS sample buffer and then analyzed by SDS-PAGE followed by western blotting with the anti-O-GlcNAc antibody (RL2). The O-GlcNAcylation competition assays were performed in 20 µl volumes with 1 µg OGT, 2 mM UDP-GlcNAc, 0.5 µg TAB1, and different doses of OGA-D175N (0.25/0.5/1 µg) in the reaction buffer (25 mM Tris pH 8.0, 100 mM NaCl, 5 mM MgCl$_2$, 1 mM DTT).

For the O-GlcNAcase assay, either human O-GlcNAcylated TAB1 (G-TAB1) or O-GlcNAcylated YTHDF1 (G-YTHDF1) was used as the substrate. The assays were performed in 20 µl volumes with 2 µg OGA and 1 µg G-TAB1/G-YTHDF1 in the reaction buffer (25 mM Tris pH 8.0, 100 mM NaCl, 5 mM MgCl$_2$, 1 mM DTT) for 90 min at 37 °C. The reaction mixtures were analyzed by SDS-PAGE and Western blotting with the anti-O-GlcNAc antibody (RL2) (Abcam, ab2739). The quantitative OGA activity assays were performed using O-GlcNAcase assay kit with PNP-GlcNAc as substrate following the manufacturer's protocol (BMR, E-130). Unpaired two-tailed $t$-test was used to calculate the p values.

### Mammalian cell culture and cellular O-GlcNAcylation assays
HEK293FT cells were cultured in 10-cm or 6-well plates in a 37 °C incubator with 5% CO$_2$ in the GIBCO$^{TM}$ DMEM (Fisher Scientific) medium supplemented with 10% fetal bovine serum (Sigma Aldrich) and 1% penicillin-streptomycin-glutamine (Invitrogen). The sgRNA targeting the C terminal region of human OGT (5'-agcataaataaagactgcac-3') was ligated into the pSpCas9[55]-2A-Puro (PX459) V2.0 vector. The homology-directed repair (HDR) template containing the 5' homology arm (-500 bp), the HaloTag9-3X Flag tag-P2A-Hygro (hygromycin B resistance gene) cassette, and the 3' homology arm (-500 bp) was cloned into the pUC19 vector and co-transfected with the Cas9 plasmid into human 293FT cells using Lipofectamine 3000 (Thermo Fisher Scientific). At 6 h after transfection, the media were replaced with fresh media containing 5 µM farrerol[56]. At 24 h after transfection, the media were changed to fresh DMEM complete media. After several rounds of hygromycin B (200 µg/ml) selection, single clones were picked and seeded into new 6-well plates. The clones were screened by PCR sequencing and Western blotting with the anti-OGT and anti-Flag antibodies for the integration of the Halo-Flag tag cassette into the endogenous OGT locus. Depletion of the resulting OGT-Halo-Flag fusion protein in 293FT cells was induced with the addition of Halo-PROTAC3 (2 µM). The cell samples were collected at different time points and analyzed by Western blotting.

The HEK293FT OGT-Halo-Flag dual tags knock-in cells were transfected with Myc-OGT WT and mutant plasmids using Lipofectamine

2000 (Thermo Fisher Scientific). The cells were then treated with HaloPROTAC3 (2 μM) in the culture media for 24 h. The cells were harvested, re-suspended in 2× SDS sample buffer, boiled for 5 min at 95 °C, and analyzed by SDS-PAGE followed by western blotting.

The primary antibodies used in western blotting included: rabbit anti-OGT (Abcam, ab177941), mouse anti-OGA (Abcam, ab68522), rabbit anti-OGA (Proteintech, 14711-1-AP), mouse anti-Myc (Sigma-Aldrich, M4439), mouse anti-Flag (Sigma-Aldrich, F1804), mouse anti-O-GlcNAc (Cell Signaling, CTD110.6, 9875 s), mouse anti-O-GlcNAc (RL2) (Abcam, ab2739), and mouse anti-GAPDH (Proteintech, 60004-1-Ig).

### Cryo-EM data collection and image processing
OGT alone and the OGT-OGA complex were incubated with 1 mg/ml UDP-GlcNAc on ice for 30 min. For cryo-EM grid preparation, 3 μl samples (~5 mg/ml) were applied onto glow-discharged holey carbon grids (Quantifoil Cu R1.2/1.3, 300 mesh), blotted with a Vitrobot Marker IV (Thermo Fisher Scientific) for 3 s under 100% humidity at 4 °C, and subjected to plunge freezing into liquid ethane. All cryo-EM data were collected using the FEI Titan Krios microscope at 300 kV equipped with a Gatan K3 Summit direct electron detector (super-resolution mode, at a nominal magnification of 105,000) and a GIF-quantum energy filter. Defocus values were set from −1.8 to −2.3 μm. Each stack of 32 frames was exposed for 2.13 s, with a total electron dose of 50 e$^-$/Å$^2$. AutoEMation was used for fully automated data collection[57].

All micrograph stacks were motion-corrected with MotionCor2[58] with a binning factor of 2, resulting in a pixel size of 0.861 Å. Contrast transfer function (CTF) parameters were estimated using Gctf[59]. Most steps of image processing were performed using cryoSPARC[60]. For 3D processing of the OGT data, a total of 6,266,880 particles were automatically picked from 5539 micrographs using Gautomatch (developed by Kai Zhang, MRC-LMB). Particles were extracted with a pixel size of 3.444 Å and subjected to several rounds of reference-free 2D classification. 1,264,325 particles were kept after the exclusion of obvious ice contamination and junk particles. Then, ab initio models were generated and subsequently used for heterogeneous 3D refinement. The best class of 493,491 particles were reextracted without binning. After the last round of 3D classification, 145,243 particles were used for further 3D refinement, including homologous refinement, heterogeneous refinement, non-uniform refinement, and local refinement. The global resolution of the OGT homodimer is 3.69 Å based on the Fourier Shell Correlation (FSC) 0.143 criterion.

For data processing of the OGT-OGA complex, 5,316,274 particles were picked using Gautomatch from 5744 micrographs. After particle extraction with a binning factor of 4, 458,373 particles were reextracted without binning after the last round of 3D classification. Upon several rounds of 2D and 3D classification combined with different subsets, three major conformations of the OGT-OGA complex emerged: conformation I (85,916 particles), conformation II (114,543 particles), and conformation III (54,789 particles). After the final round of 3D refinement, the global resolutions of the three conformations were determined to be 5.68 Å, 3.92 Å, and 5.86 Å, respectively, using FSC 0.143 criterion. Finally, cryo-EM density maps were sharpened using the negative B-factor reported by cryoSPARC[60]. Conformation II of the OGT−OGA complex had the highest resolution and was used for further structural analysis.

For data processing of human OGT monomeric W208A/L209A/I211A/H212A (4A) mutant and human OGA protein, 135,924 and 271,000 particles were used for the final round of 2D classification, respectively.

### Model building and refinement
The X-ray structures of human OGT (PDB: 1W3B and 3PE3) or OGA (PDB: 5UN9) were used as the starting models and docked into the final EM maps with UCSF Chimera[61]. The models were manually adjusted and iteratively built in COOT[62] and then refined against summed maps using phenix.real_space_refine implemented in PHENIX[63] until the validation data were reasonable. FSC values were calculated between the resulting models and the two half-maps, as well as the averaged map of the two half-maps. The quality of the models was evaluated with MolProbity[64] and EMRinger[65]. The structure validation statistics were listed in Supplementary Table 1. All structural figures were prepared with PyMOL[66], Chimera[61] or Chimera X[67].

### Statistical analysis
No statistical methods were used to predetermine sample size or applied to data analysis. The experiments were not randomized. The investigators were not blinded to allocation during experiments and outcome assessment.

### Reporting summary
Further information on research design is available in the Nature Portfolio Reporting Summary linked to this article.

## Data availability
The cryo-EM density maps of the OGT dimer and the OGT-OGA complex generated in this study have been deposited to the Electron Microscopy Data Bank under the accession numbers EMD-33768 (OGT dimer), EMD-33767 (OGT-OGA conformer I), EMD-33773 (OGT-OGA conformer II), and EMD-33769 (OGT-OGA conformer III). Atomic coordinates have been deposited to the RCSB Protein Data Bank under the accession numbers 7YEA (OGT dimer) and 7YEH (OGT-OGA conformer II). Source data are provided with this paper.

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

## Acknowledgements

Single particle cryo-EM data were collected at the Westlake University Cryo-EM Facility. We thank Jianning Lei, Li Huang, Zhipeng Jiang, and Xiaojuan Wang for technical support and facility access. We thank the Westlake University High-Performance Computing Center for computational resources and technical assistance. This work was supported by the National Natural Science Foundation of China (Project 32271258 to H.G., Project 32130053 to H.Y.), and New Cornerstone Investigator Program (to H.Y.).

## Author contributions

H.G. initiated the project and designed the experiments. H.G. and P.L. purified the OGT, OGA, the OGT–OGA complex, and TAB1 proteins; performed negative-stain electron microscopy, cryo-EM grid preparation, data collection and processing, and model building; performed glycosylation and binding assays. P.L. generated the OGT-HaloTag9 knock-in cell line and performed cellular assays with assistance from S.Q.. Y.L. and M.H. participated in cryo-EM grid preparation and data collection. T.C. and M.Y. participated in cryo-EM data collection. H.Y. and H.G. supervised the project. H.Y. and H.G. analyzed the data and wrote the manuscript with input from all authors.

## Competing interests

The authors declare no competing interests.
