## [Peer Review File · Nature Communications]

Cryo-EM structure of human O-GlcNAcylation enzyme pair
OGT-OGA complexREVIEWER COMMENTS

Reviewer #1 (Remarks to the Author):

See attached word document

OGT and OGA are a pair of enzymes solely responsible for O-GlcNAcylation in eukaryotes. Previous studies have focused on the structures and catalytic mechanisms of the individual enzymes. Two major questions in the field are how a single pair of enzymes can recognize and act on several thousands of protein substrates, and how the two opposing enzymes regulate each other. This paper by Lu et al. provides insights to both questions, and as such, is an important contribution to the field. The key novelty of the work is the *in vitro* assembly of the OGT-OGA complex and the determination of the complex structure at 3.9 Å resolution. The structure reveals a mutual inhibition: OGA occupies the substrate binding groove and the catalytic pocket to inhibit the OGT, and the OGT dimer disrupt the OGA dimer interface that is required for OGA activity, thereby inhibiting the OGA. The structural observations are complemented by biochemical assays using the catalytically inactive mutants (hOGA-D175N and hOGT-H508A) as controls. The structure and functional assays are generally well done, and this reviewer considers the work a breakthrough, adding significantly to our understanding of the crosstalk between OGT and OGA and the maintenance of O-GlcNAc homeostasis. A few concerns listed below should be addressed before it is suitable for publication.

Major concerns

1. The authors suggest their structure is a product complex in which OGA Ser-405 may have been modified with GlcNAc by OGT. Please show the EM density around the catalytic pocket, whether there is enough resolution to convincingly conclude GlcNAc is covalently linked to S405, and whether there is a density gap between UDP and GlcNAc to show that UDP-GlcNAc hydrolysis has occurred. Because the overall resolution is relatively low, these key linkages may not be clearly established by experimental density. If true, the author should consider increasing the size of the dataset or performing a focused refinement to improve the local density. A mass spec analysis should be considered to confirm the proposed Ser-405 modification.
2. I wonder if the authors are a bit overselling their work. In Discussion, it is important to point out the two enzymes are likely to exist in both individual enzyme form and the complex form. It is currently unclear the portions of these forms, when and where the complex forms, and the regulatory mechanism for complex dissolution. In Fig S5a, the gel filtration peak is lopsided indicative of an equilibrium of complex and the individual enzymes. In Fig 1c Halo-PROTAC co-depletion assay, there is a nearly 10-hour delay in OGA reduction vs OGT reduction, suggesting that a majority of cellular OGT is not bound to OGA. This should be discussed as well.

3. The OGT dimer in the OGT-OGA complex appears symmetric. If so, a second OGA monomer should be able to bind; if not, please explain if there is conflict(s) preventing the binding of a second copy of the monomeric OGA.

Minor concerns

1. Page 5 Line 5 from bottom: a lumen of 22 Å in diameter – label this in Fig. 1d.
2. Page 6 Line 4 from bottom: ~80 Å should be labeled in Fig.
3. Page 7 Line 3: Clarify if the OGA dimerization interface is involved in OGT binding.
4. Page 7 Lines 3-5 from bottom: Labeled the mentioned H bond, label the described residue A664 in figure panel(s).
5. Page 8 Line 4: GlcNAc attachment to S405 – see major comment #1.
6. Page 9 Line 3-4 from bottom: Provide an explanation or speculation why the product complex is stable.
7. Page 11, Lines 8-9: spell out the identities of the catalytic residues of OGA.
8. Page 13 Line 6: “straightening” I wonder if “partial uncoiling” is easier for reader.
9. Page 13 Line 8 from bottom: 2 typos, “solenoid”.
10. Page 14 Line 6 from bottom: “only substrates that can outcompete ---” This is likely an overstatement, as OGT dimer very likely also exist as a subpopulation in vivo.
11. In Fig. 5b, OGT should be drawn as a dimer to be more accurate.
12. In Fig. 2g legend: residues 371-440, not 371-44.
13. In Fig. S1b last lane: the weak OGT band indicating presence or use of little mutant enzyme, yet a faint product band is detected, conflicting with the assertion H508A is inactive. Rerun the experiment use equal amount of proteins.
14. Fig. S2C: Flowchart – indicate the applied symmetry C1 or C2; in the local resolution map panel, the scale bar is too small and illegible.
15. Fig. S2D: X-axis is in Fourier (reciprocal) space, label with $1/(14 \text{ \AA})$, $1/(6.9 \text{ \AA})$ etc..
16. Fig. S6c – legend is missing.

Reviewer #2 (Remarks to the Author):

In this work, Lu et al provide a model of the O-GlcNAc cycling enzymes working together to inhibit futile cycling and improve selectivity of O-GlcNAc modification. The Cryo-EM approach coupled with the in vitro assays and mutagenesis provides critical support for their model (Fig 5). There are a number of minor issues but overall enthusiasm is high for this manuscript that will be of high interest to those studying not only the O-GlcNAc modification but any post-translational modification capable of cycling.

Minor issues:

1. A thorough editing of the manuscript for word choice and grammar would improve clarity.
2. References and methods are mixed together in the PDF document (Refs 1-34 and 35-45 are split).
3. UDP-GlcNAc should be referred to as a substrate donor or a sugar nucleotide substrate donor, NOT as a cofactor.
4. Given that mutations in the OGT gene lead to X-linked intellectual disability (PMID: 28302723), it is surprising that this isn't at least mentioned in the introduction OR in the discussion. Addressing where the >10 missense variants linked to XLID in the structure of OGT are and potential impact would add significance.
5. A more detailed comparison of their structure to the existing Cryo-EM and X-ray structures of OGT and OGA...especially focusing on any differences (there were some important differences with the X-ray structure and recent Cryo-EM structure of OGT--does your data support the recently observed Cryo-EM differences?)
6. The final paragraph while important is speculation on a very interesting proposed model. It should be made clear...something along the lines of "our data supports a working model in which..."...also, what future work could further test this model?
7. Ref. 31 is biorxiv for 2-3 years without being published in peer reviewed literature...perhaps a reference to the Zachara laboratory that has used Cre/Lox KO cell lines for sometime now would be more appropriate.

Reviewer #3 (Remarks to the Author):

In this manuscript the authors use cryo-EM to model the structure of OGT as well as OGT with a known substrate: OGA. The experiments described in this manuscript could potentially give insights into OGT structure, substrate recognition and catalysis. The authors also make claims that the OGT/OGA complex acts in vivo to sterically inhibit the other enzymes and is critical for maintaining O-GlcNAc homeostasis. However, the authors do not satisfactorily test or have data support these conclusions. The major flaws

with the manuscript center around biochemical characterization of the complex formed and its existence in situ.

Major issues with specific figures and further suggested experiments are outlined below.

1. Gels/Blots presented in this manuscript are cut-off and do not contain a ladder to assess size. Entire gels/blots should be provided (can be in extended figures) and ladders for gels/blots should be included in all figures. These are necessary for interpretation. Further all experiments should be performed multiple times, quantified and statistically analyzed.

2. Figure 1C. what is the antibody being used to detect OGT? Is it against OGT or the Halo tag? The author's interpreted the decrease in OGA to indicate that OGT acts to stabilize OGA protein, however, there is no evidence of this.

3. From figure 1C the authors conclude in lines 104-107 that "these findings suggest that large pools of OGT and OGA might form stable complexes in human cells and depend on each other for stability" However, the authors do not perform any experiment to test this conclusion. The author's need to assess OGA transcription by qRT-PCR at the various time points to show that the effect is not due to coordinate transcriptional regulation. Further to prove that the loss of OGA is due to protein stability the author's need to inhibit the proteasome to show that OGA levels remain. Additionally, the authors never show that OGT and OGA actually form a complex in vivo, the authors need to do co-IPs in cells not just in-vitro to show that a stable complex is formed. Assessment of in vivo binding of OGT and OGA should also be performed under varying nutrient conditions that alter the level of UDP-GlcNAc as the authors conclude that this complex forms to inhibit O-GlcNAcylation under conditions of high UDP-GlcNAc.

4. All OGT activity assays should be repeated using additional substrates as it is difficult to make claims on the overall activity of OGT in a cell based on glycosylation of 1 substrate in vitro. Other good substrates include CKII or Nucleoporin 62.

5. Throughout the paper the authors use a glycosylated-TAB1 for experiments. However, this TAB1 has multiple bands, of which are lower molecular weight than TAB1 as can be seen in their blots. In figure 4F the authors state that the increase in lower bands when incubated with OGA was due to de-glycosylation, however, as can be seen in the other figures glycosylated-TAB1 (when incubating TAB1 with OGT) runs the same as TAB1. It is unclear what all of these bands are. The authors need to provide a better test of OGA activity; please see PMID: 28472822 (Kim, Eun Ju., *Chembiochem*. 2017) for a review on quantitative OGA activity assays

6. The authors state that OGA acts to sterically inhibit OGT, however, this is never tested using a competition assay.

7. The authors state that after OGT glycosylates OGA it remains in a complex (lines 174-175), this would suggest that OGA needs to be glycosylated in order to form a stable complex, the authors should test whether the catalytic mutant of OGT forms a stable complex with OGA.

8. Figure 2G and extended figure 8B gel lanes are inappropriately cut off. Additionally, as mentioned above pull downs should be done by overexpression in cells not in-vitro.
9. In extended figure 8C. Why is there no endogenous OGT/OGA and very little O-GlcNAcylation in the cells without HaloPROTAC?
10. In figure 8D. What is being used for analysis? Entire lane? Specific band? What statistics are used?
11. Extended figure 9 and figure 1F show inconsistent results for the ability of OGT4A to glycosylate. Please repeat all experiments multiple times and quantify/use appropriate statistics.
12. Throughout the paper OGT4A consistently runs higher than WTOGT or OGTH508A. However, in figure 3D, lanes 6-8 the WT or OGT508A is running higher than WT/HOGT508A has been (as can be seen when compared to lanes 2-4) and similar to OGT4A. Was OGT4A accidentally used in these lanes? If so, this figure is not interpretable.
13. As mentioned above; the authors have no data to support the model in figure 5B or in their dis

In summary, the findings have potential implications for understanding O-GlcNAc homeostasis but fall short in several significant ways. While some insight is gained from looking at the complex formed in vitro, the extrapolation to in vivo significance is problematic.

Reviewer #4 (Remarks to the Author):

This manuscript reports Cryo-EM structures of OGT and OGT/OGA complexes as well as biochemical studies that validate structural findings or explore possible implications of the structures. Given the importance of OGT, OGA, and O-GlcNAc cycling in mammalian cell biology (and metazoan biology in general), the structures reported are an important contribution to the field and justify publishing the manuscript. However, before it can be published, the authors need to make revisions to the manuscript to address the following comments.

Major comments:

1. The authors draw a number of conclusions that are unwarranted based on the data presented. They should either carry out additional experiments to support currently unwarranted conclusions or modify the text. As appropriate, they should also acknowledge other interpretations of experimental results. See below for specific issues that need addressing.

- a) The authors conclude that OGT stabilizes OGA in cells. This conclusion is based on two lines of evidence: 1) OGT forms a complex with OGA. 2) When OGT is degraded in cells, OGA becomes depleted.

Because O-GlcNAc levels also decrease when OGT is degraded, and low O-GlcNAc levels are known to result in reduced OGA levels, the authors cannot conclude from their data that OGT directly stabilizes OGA. They will need to carry out additional experiments to draw this conclusion. Alternatively, they can speculate that OGT might stabilize OGA directly – but will need to acknowledge that there are other mechanisms by which OGT/O-GlcNAc regulate OGA expression. For example, inhibiting OGT leads to reduced OGA by mechanisms that include O-GlcNAc-dependent transcriptional control (transcription and splicing mechanisms - see 10.3389/fendo.2014.00206 and 10.1093/nar/gkaa263). OGT levels are also regulated by OGA via transcriptional mechanisms (that do not rely on OGT/OGA complex formation), and by splicing mechanisms that are dependent on O-GlcNAc levels (see 10.1074/jbc.RA118.004709 and 10.1016/j.celrep.2017.07.01, respectively). Reference 32 cited in the manuscript also identifies additional mechanisms for regulation of OGT and OGA levels. Given the flimsy support for a physical stabilization model, the authors should remove from the abstract assertions that OGT directly stabilizes OGA.

b) The authors speculate that OGT and OGA form pools of complexes in cells that have a mutually regulatory effect. Consistent with this, cellular OGT can pull down OGA. However, OGT has many other interaction partners, and some pull down more reliably with OGT than OGA does (e.g., HCF-1, TRAK1). It is premature to propose that OGA negatively regulates OGT activity by binding to OGT when other proteins also bind to OGT, especially given that there is no quantitative information about how tightly OGA and various other proteins bind to OGT. The authors either need to do more experiments (including in cells) to support their model for OGA regulation of OGT activity, or they need to revise the manuscript to reduce emphasis on OGA's role in negative regulation of OGT.

c) Regarding the model of mutual regulation through physical interaction: it is worth noting here that the authors' model would predict that OGA and OGT are colocalized in cells. They are commonly found in the same compartment, but how much of one colocalizes with the other has not been rigorously investigated (to my knowledge). However, there is one subcellular location where OGA is present and OGT is reported to be excluded: nucleoli. This observation undercuts the generality of their (speculative) model. I would urge the authors to exercise greater caution and not overinterpret the implications of their structures.

d) Regarding the topological model in which the OGT TPR superhelix unwinds to accommodate entry of substrates: maybe unwinding occurs – that is one solution for how substrates with disordered regions sandwiched between folded domains can enter the lumen. However, there is no support for the unwinding model (as the authors acknowledge). The authors should soften the language asserting that a major unwinding of the TPR domain is required for substrate entry. In addition, the model appears to assume that substrates interact with the full length of the TPR lumen, but so far only a very small number of substrates have been investigated for their dependency on the TPR domain – and the results suggest that not all substrates bind to the full TPR domain.

e) The authors also propose that unwinding is required to release substrates, and they further propose that enclosing substrates in the TPR lumen allows processive glycosylation to occur. An interesting thing about OGT is that it often glycosylates only one or two ser/thr residues in long polypeptides in which many such residues are present. So for what substrates are the authors proposing a processive model? They should clarify this point because the text seems to suggest that OGT processively glycosylates a lot of substrates. (OGT was suggested to processively glycosylate the CTD of RNA Pol II, which is heavily glycosylated. However, an investigation showed that glycosylation of this substrate is distributive. See 10.1021/acs.biochem.5b01280.) If the authors are unable to identify multiple references that contain experiments demonstrating processive glycosylation of substrates by OGT, they should remove this line of speculation.

2. I don't understand a key aspect of the model in Fig. 5b. Can the authors explain more clearly how high O-GlcNAc levels allow OGT substrates to outcompete OGA for binding to OGT? What might be the physical mechanisms that allow this to occur?

Related to the model: the authors suggest that OGA negatively regulates OGT by binding to it stably enough to prevent glycosylation of 'weak' potential substrates; however, they propose that substrates that bind more tightly than OGA can displace OGA and be glycosylated. But if these good substrates bind tightly enough to OGT to displace something that binds tightly enough to pull down with OGT, wouldn't that present a problem for turnover? I might be missing something, but the suggested model doesn't completely make sense.

Related to testing the model: I understand the utility of picking a single substrate to use in experiments, but the authors should not make broad claims based on one substrate. The authors use TAB1 throughout the manuscript as a substrate for their in vitro biochemistry. But not all substrates are equal – see Joiner et al. (10.1021/acs.biochem.0c00981) and Shen et al. (10.1074/jbc.M111.310664). The authors need to tone down all claims that rely on using a single substrate as the 'test pool' for glycosylation activity.

3. Another problem with the manuscript is that the authors do not clearly put their findings in the context of other work. In several places, they seem to claim credit for findings that are very similar to findings of others, but without attribution. Some examples:

a) The authors make a dimer-disrupting mutant of OGT and report that the mutant is less active at glycosylating the substrate TAB1 than wildtype OGT. Previously, Davies and coworkers, who published the first cryoEM structure of OGT (Meek et al. ref 28 in the manuscript), reported that a dimer-disrupting OGT mutant was less active at glycosylating TAB1. Because the results here are the same as those in the Meek et al. paper, this TAB1 experiment should not be included in the manuscript – certainly not as a main text figure panel. The minor modifications in experimental design (e.g., these

authors use a different dimer-disrupting mutant and assay glycosylation differently than did Davies and coworkers) do not justify including Figure 1f. The authors should move it to the SI, note that their results agree with previous studies, and cite the previous work in discussing effects of the dimer. In doing so, the authors should take care to be accurate. Currently, they state, “OGT functions as a homodimer” (lines 99-100); however, data in the Meek et al. paper show that substrates that extend far up into the lumen are sensitive to dimer status (TAB1), but shorter substrates (one based on an HCF-1pro repeat) are not.

b) Asparagine ladder mutants were previously shown to have attenuated glycosylation activity in vitro and in cells. The authors should appropriately reference earlier examples of work reporting attenuated glycosylation activity for asparagine ladder mutants in vitro and in cells. Levine et al. reported in vitro studies showing that a five asparagine ladder mutant showed reduced glycosylation of protein substrates in cell extracts (10.1021/jacs.7b13546) and also reported attenuated glycosylation activity for the five asparagine mutant in cells (10.1073/pnas.2016778118). Other asparagine ladder mutants were also reported to have attenuated glycosylation activity using both purified protein substrates, including TAB1, and cell extracts (10.1021/acs.biochem.0c00981). And Kositzke et al. (ref. 29) studied interactions of OGA with individual luminal asparagine mutants, but this is not mentioned even though the reference is cited in a different context.

c) The authors should specifically comment on whether their OGA results agree or not with findings of Kositzke et al., who similarly reported that Asns in different positions have different effects on OGA binding (Kositzke et al. used crosslinking as a proxy for binding affinity, but the comparison should still be made).

d) The authors note that asparagine ladder residues contact the OGA backbone to anchor it in the TPR lumen. A crystal structure reported in 2013 showed that asparagine ladder residues can anchor peptides in the TPR lumen by making backbone contacts (10.1126/science.1243990). This was the first structure showing a peptide (HCF-1pro) bound in the TPR lumen and should be cited where the authors describe how OGA’s disordered region binds to the TPR domain of OGT because it informed understanding of luminal binding. (And actually, it included the first in vitro experiments on Asn ladder mutants.) Are similar contacts made to the peptide backbone of HCF1pro as to OGA in the part of the lumen that both structures contain? (After correcting for the ten residue offset in the sequence numbering, it looks like several of the asparagine backbone contacts are similar.) A subsequent structural study fused part of the disordered region of TAB1 to OGT to obtain a structure showing contacts to the TPR lumen (10.1098/rsob.170078). A comment might be warranted if the contacts are consistent – although because this is a covalent fusion with a TPR-truncated OGT, the positioning may be biased.

With the previous and the current structures showing different peptides bound in the TPR lumen, it is reasonable for the authors to make generalizations for how the asparagine ladder anchors peptides in the lumen. A comparison of OGT contacts to side chains of OGA and the HCFpro peptide used in the

previous study might also allow some statements to be made about how selectivity is achieved. But the authors need to put their findings in context because other papers (cited above) have specifically described the role of asparagine ladder residues in backbone anchoring and have also identified other side chains as important in selectivity (via side chain contacts). It does not detract from the value of the structures reported in this manuscript to acknowledge earlier work that focused on how substrates bind in the TPR lumen and reached broadly similar conclusions. Indeed, appropriate comparisons would likely enrich their analysis.

4. Rigor/technical issues/questions:

a) As noted above, the authors show that depletion of Halo-OGT results in depletion of OGA and conclude that OGT-OGA complexation stabilizes OGA. To evaluate whether OGT itself or O-GlcNAc regulates OGA abundance, the authors should consider providing a similar time course monitoring OGA levels when O-GlcNAc is lost through inhibition of OGT. A very good inhibitor of OGT exists (10.1021/jacs.8b07328) and can be purchased (OSMI-4). It would be useful to know the extent to which OGA levels decrease upon OGT active site inhibition, which results in decreased O-GlcNAc -which is known to affect OGA expression and functional mRNA abundance. (That said, a confounding factor could be that inhibitor binding in the active site displaces OGA, which would complicate interpretation of the results. The authors would need to look at this.) Alternatively, or in addition, OGA levels should be monitored when a catalytic dead variant of OGT is put into cells and Halo-OGT is degraded. Replicates and quantitation need to be performed. But if the authors really want to confirm their model, they should perform pulse-chain experiments that monitor OGA degradation rates under normal conditions and when OGT is depleted, and should provide some quantitation/estimate of OGT-OGA complex pools in cells. Otherwise, they should alter the text.

b) Extended Data Fig. 1a: what is the gel below the gel filtration profiles supposed to show? None of the lanes are labeled and the caption does not describe it.

c) Extended Data Fig. 1b: These data are described as showing that the H508A mutant does not glycosylate TAB1 at all. However, the amount of H508A used in the experiments appears to be ~25% the amount of WT OGT, which complicates interpretation. Moreover, there appears to be signal in the last lane of the WB (there is a disruption in the band that could be due to a transfer artifact). This experiment should be repeated using similar levels of WT and H508A mutant and replicates should be performed. (It is worth pointing out that the same mutant was tested in 10.1126/science.1243990 and shown to have some activity for glycosylating the CKII peptide. See SI – H498A mutant corresponds to the H508A mutant. The mutant does not appear to be fully catalytically dead. K852M, used later in the paper, is a more reliable catalytic dead mutant. Also, molecular weights should be indicated on the gels/blots.

d) The western blot quality in Fig. 3d is poor and this experiment needs to be repeated. It is not possible to tell if there is faint signal in other lanes (than WT) because the background is so dark.

e) Fig. 4d: why are there such different amounts of OGT present in lanes containing OGT? Why does the G-TAB1 CBB signal look so different in lane 6 (from the left)?

f) Extended data Fig. 8c/d: how were the relative O-GlcNAc levels quantified? Did the authors correct for Myc-OGT levels? (And can OGT and Myc-OGT be distinguished by size?) More details on how quantitation was performed are required to understand the experiment. Also, the y-axis in the plot (8d) is misleading because the three identical divisions represent three different quantities (0.6, 0.2, and 0.4). The authors should make a proportionate linear scale from 0 and annotate at regular intervals. Data for the three independent replicates used to generate the plot should also be shown. The authors should clarify if independent replicates means different transfections.

g) Extended data Fig. 9: How was the relative quantitation performed? (I assume the numbers below the WB represent relative quantitation results.) Are there replicates for this experiment? Replicates should be performed and quantitation should include errors.

h) Re: extended data Fig. 9: there is a mobility difference between WT OGT and all the mutants, regardless of the location of mutations. Can the authors explain the apparent difference in size?

i) Why did the authors make the 4A dimer-disrupting mutant instead of using the mutant others have reported in the literature (W198E/I201D)?

j) Why do OGA levels increase out to 9 hours as OGT and O-GlcNAc are depleted? Is the blot in Fig. 1c representative? If this is reproducible, how do the results fit with the proposed model?

k) Can the authors provide confirmatory evidence that Halo tag knock-in hit only the OGT locus?

l) The description of complex preparation does not mention UDP-GlcNAc. Did the authors include UDP-GlcNAc in setting up the complexes, which show UDP and glycopeptide? Please clarify.

m) For synthetic human genes, were nucleotide sequences codon-optimized for bacterial expression? The authors should clarify which genes were cloned from a human cDNA library and which were

synthesized. Are the encoded proteins otherwise identical to the human proteins? The authors should include full CBB gels and WBs showing purified proteins.

n) The authors used structure 3PE3 in model building, but their structure contains a glycopeptide. There are deposited structures with glycopeptides and thioglycopeptides (e.g., I think the first were 4GYw, 4GZ3). Any reason these were not used?

o) The authors cite Lazarus et al. Nature 2011 when talking about glycopeptides but the structures in this paper did not include UDP-sugars or glycopeptides bound. A subsequent Lazarus et al. paper included those structures ((2012) Nat Chem Biol 8: 966-968).

n) The manuscript is poorly referenced. Citations to previous work on the asparagine ladder (see above) are lacking, but the problem is more pervasive. For example, in lines 225-227, the authors write, "Different subsets of TPRs are required for the recognition of different substrates..." but do not provide any citations. Another example of a sentence where a citation is required (line 229-30): "OGT prefers to bind to and act on a long segment (at least 50 residues) of intrinsically disordered regions in substrates." And line 64-65: "Biochemical and structural studies on human OGT and OGA have provided critical insights into the mechanistic [sic] underlying their enzymatic activities towards peptide substrates." There are other examples of sentences that need citations and the authors should go through the manuscript and provide them.

Other:

--There are a fair number of typos in the manuscript and the authors should go over it more carefully and correct errors in main text, methods, and extended data.

--The authors refer to the "asparagine ladder" within the OGT TPR lumen in talking about asparagines that face into the lumen. Canonically, the asparagine ladder comprises the conserved Asn residues at the 6th position within each TPR (some of the refs cited above describe the Asn ladder). The authors should clarify which backbone contacts are from conserved Asn ladder residues and which are from other Asn residues. A comment on the conservation of other residues that contact the OGA backbone could be useful.

To sum up: The structure of the OGT-OGA complex justifies publishing this manuscript in a high-impact journal. There are novel aspects to the structure of the complex, including that it is the first structure showing a polypeptide bound throughout the full length of OGT's TPR lumen. The structure and accompanying biochemical experiments also imply that at least some of the time OGT and OGA are

found together in inactive complexes, suggesting a regulatory role for complex formation - although the data supporting the proposed regulatory mechanism are inconclusive (and the model in 5b for how regulation is achieved is vague). In any case, substantial revisions are required to: 1) acknowledge/discuss previous work in the field that connects to the authors' findings; 2) tone down unsupported models/conclusions; 3) provide additional support for key conclusions; 4) address technical issues; 5) provide adequate citations.

Point-by-point response to reviewers' comments

Manuscript ID: NCOMMS-22-50787-T

Previous Title: Mutual regulation mechanism of the O-GlcNAcylation enzyme pair revealed by Cryo-EM structure of human OGT–OGA complex

Current Title: Mutual regulation of the O-GlcNAcylation enzyme pair revealed by cryo-EM structure of human OGT–OGA complex

We thank the reviewers for their overall positive assessment of the quality and importance of our study. Their constructive comments and suggestions are very helpful. In response to their comments, we have performed additional experiments and revised the text and figures. As a result, the paper has been greatly improved. Our point-by-point responses to the reviewers follow. The reviewer's comments are in black and our responses are highlighted in blue.

Reviewer #1 (Remarks to the Author)

OGT and OGA are a pair of enzymes solely responsible for O-GlcNAcylation in eukaryotes. Previous studies have focused on the structures and catalytic mechanisms of the individual enzymes. Two major questions in the field are how a single pair of enzymes can recognize and act on several thousands of protein substrates, and how the two opposing enzymes regulate each other. This paper by Lu et al. provides insights to both questions, and as such, is an important contribution to the field. The key novelty of the work is the in vitro assembly of the OGT-OGA complex and the determination of the complex structure at 3.9 Å resolution. The structure reveals a mutual inhibition: OGA occupies the substrate binding groove and the catalytic pocket to inhibit the OGT, and the OGT dimer disrupt the OGA dimer interface that is required for OGA activity, thereby inhibiting the OGA. The structural observations are complemented by biochemical assays using the catalytically inactive mutants (hOGA-D175N and hOGT- H508A) as controls. The structure and functional assays are generally well done, and this reviewer considers the work a breakthrough, adding significantly to our understanding of the crosstalk between OGT and OGA and the maintenance of O-GlcNAc homeostasis. A few concerns listed below should be addressed before it is suitable for publication.

Response: We thank the reviewer for the positive comments on our work.

Major concerns

1. The authors suggest their structure is a product complex in which OGA Ser-405 may have been modified with GlcNAc by OGT. Please show the EM density around the catalytic pocket, whether there is enough resolution to convincingly conclude GlcNAc is covalently linked to S405, and whether there is a density gap between UDP and GlcNAc to show that UDP-GlcNAc hydrolysis has occurred. Because the overall resolution is relatively low, these key linkages may not be clearly established by experimental density. If true, the author should consider increasing the size of the dataset or performing a focused refinement to improve the local density. A mass spec analysis should be considered to confirm the proposed Ser-405 modification.

Response: We thank the reviewer for these constructive suggestions. Although the cryo-EM structure of the OGT–OGA complex was determined at an overall 3.9 Å resolution, it is sufficient to distinguish the local EM density for the OGT catalytic pocket and to model OGA S405 O-GlcNAc. We have added a figure to show the density of the O-GlcNAcylated S405 from OGA (Figure 1i), which clearly shows the continuous density between S405 and NAG. The density map also shows that there is a gap between the densities of UDP and GlcNAc, suggesting that UDP-GlcNAc hydrolysis has occurred. As suggested, we have performed mass spectrometry analysis of the modified OGA. Unfortunately, for unknown reasons, our analysis did not cover the tryptic peptide containing S405. On the other hand, the S405 site has previously been identified as being modified by O-GlcNAcylation using more sophisticated proteomics approaches (e.g. Khidekel et al. Probing the dynamics of O-GlcNAc glycosylation in the brain using quantitative proteomics. *Nat. Chem. Biol.* 2007, 3:339-48. doi: 10.1038/nchembio881.).

2. I wonder if the authors are a bit overselling their work. In Discussion, it is important to point out the two enzymes are likely to exist in both individual enzyme form and the complex form. It is currently unclear the portions of these forms, when and where the complex forms, and the regulatory mechanism for complex dissolution. In Fig S5a, the gel filtration peak is lopsided indicative of an equilibrium of complex and the individual enzymes. In Fig 1c Halo-PROTAC co-depletion assay, there is a nearly 10-hour delay in OGA reduction vs OGT reduction, suggesting that a majority of cellular OGT is not bound to OGA. This should be discussed as well.

Response: We agree with the reviewer that OGT and OGA likely exist in both free and complexed forms *in vivo*, depending on their expression levels and subcellular localization. We have revised the text accordingly to reflect this point. Our conclusion about mutual regulation only applies to the stable OGT–OGA complex, whose abundance *in vivo* has not been quantified and might be regulated by multiple factors.

3. The OGT dimer in the OGT-OGA complex appears symmetric. If so, a second OGA monomer should be able to bind; if not, please explain if there is conflict preventing the binding of a second copy of the monomeric OGA.

Response: We agree with the reviewer and believe that the symmetric OGT dimer could accommodate a second copy of OGA. The fact that we can only model one OGA molecule is likely due to the dynamic nature of the complex. In fact, as shown in Supplementary Fig. 5d, we observed two additional conformers during our cryo-EM analyses: Conformer I, the symmetric OGT dimer bound to two OGA monomers; Conformer III, the symmetric OGT dimer bound to two OGA IDRs). Compared to Conformer II which we focused on in the manuscript, the maps of the other two conformers were of lower resolutions: 5.68 Å and 5.86 Å, respectively. We thus did not further build the atomic models for these two maps. The multiple conformers of the OGT–OGA complex indicate that the OGT–OGA complex is indeed dynamic.

Minor concerns

1. Page 5 Line 5 from bottom: a lumen of 22 Å in diameter – label this in Fig. 1d.

Response: We have added the label in Fig. 1d as suggested.

2. Page 6 Line 4 from bottom: $\sim 80 \text{ \AA}$ should be labeled in Fig. 1.

Response: We have added the label in Fig. 1d and 1g as suggested.

3. Page 7 Line 3: Clarify the OGA dimerization interface is involved in OGT binding.

Response: Thanks for the suggestion. In the revised manuscript, we have rewritten the sentence as “Interestingly, the domain-swapped OGA dimer is disrupted in the OGT–OGA complex, indicating that the OGA dimerization interface is involved in OGT binding.”

4. Page 7 Lines 3-5 from bottom: Labeled the mentioned H bond, label the described residue A664 in figure panel(s).

Response: In the revised manuscript, we have shown residue A664 in Supplementary Fig. 6c and linked O-GlcNAc to OGA S405.

5. Page 8 Line 4: GlcNAc attachment to S405 – see major comment #1.

Response: In addition to the previous Supplementary Fig. 6c to show the density of OGA S405, we have added a figure (Fig. 1i) to show the cryo-EM density of the O-GlcNAcylated S405 from OGA.

6. Page 9 Line 3-4 from bottom: Provide an explanation or speculation why the product complex is stable.

Response: We speculate that the enzyme-product complex is stable due to its slow dissociation rate. We have mentioned this point in our revised manuscript.

7. Page 11, Lines 8-9: spell out the identities of the catalytic residues of OGA.

Response: In our revised manuscript, we have revised the sentence to include the identities of the catalytic residues (Asp174 and Asp175) of OGA.

8. Page 13 Line 6: “straightening” I wonder if “partial uncoiling” is easier for reader.

Response: As suggested, we have changed “straightening” to “partial uncoiling” in our revised manuscript.

9. Page 13 Line 8 from bottom: 2 typos, “solenoid”.

Response: We have corrected these two typos in our revised manuscript.

10. Page 14 Line 6 from bottom: “only substrates that can outcompete ---” This is likely an overstatement, as OGT dimer very likely also exist as a subpopulation in vivo.

Response: We agree with the reviewer and have revised this statement.

11. In Fig. 5b, OGT should be drawn as a dimer to be more accurate.

Response: We have revised the figure as suggested.

12. In Fig. 2g legend: residues 371-440, not 371-44.

Response: This typo has been corrected.

13. In Fig. S1b last lane: the weak OGT band indicating presence or use of little mutant enzyme, yet a faint product band is detected, conflicting with the assertion H508A is inactive. Rerun the experiment use equal amount of proteins.

Response: As suggested by Reviewer #4, we have now used the OGT K852M mutant (a more reliable catalytic dead mutant) as the negative control (Supplementary Fig. 1d). Equal amounts of WT and mutant proteins are loaded.

14. Fig. S2C: Flowchart – indicate the applied symmetry C1 or C2; in the local resolution map panel, the scale bar is too small and illegible.

Response: In our revised manuscript, we have added the information on the applied symmetry and provided a bigger scale bar in the local resolution map panel.

15. Fig. S2D: X-axis is in Fourier (reciprocal) space, label with $1/(14 \text{ \AA})$, $1/(6.9 \text{ \AA})$ etc.

Response: Fig. S2d shows the standard resolution curve by CryoSPARC. The axis labeling follows normal practice and is widely used and accepted by the cryo-EM field.

16. Fig. S6c – legend is missing.

Response: We have added the figure legend for this figure in our revised manuscript.

Reviewer #2 (Remarks to the Author):

In this work, Lu et al provide a model of the O-GlcNAc cycling enzymes working together to inhibit futile cycling and improve selectivity of O-GlcNAc modification. The Cryo-EM approach coupled with the in vitro assays and mutagenesis provides critical support for their model (Fig 5). There are a number of minor issues, but overall enthusiasm is high for this manuscript that will be of high interest to those studying not only the O-GlcNAc modification but any post-translational modification capable of cycling.

Response: We thank the reviewer for the overall positive assessment of our work.

Minor issues:

1. A thorough editing of the manuscript for word choice and grammar would improve clarity.

Response: We have carefully edited the text and corrected some typos in the revised manuscript.

2. *References and methods are mixed together in the PDF document (Refs 1-34 and 35-45 are split).*

Response: We have corrected this issue in our revised manuscript.

3. *UDP-GlcNAc should be referred to as a substrate donor or a sugar nucleotide substrate donor, NOT as a cofactor.*

Response: We have made this change throughout the text as suggested.

4. *Given that mutations in the OGT gene lead to X-linked intellectual disability (PMID: 28302723), it is surprising that this isn't at least mentioned in the introduction OR in the discussion. Addressing where the >10 missense variants linked to XLID in the structure of OGT are and potential impact would add significance.*

Response: We thank the reviewer for this great suggestion. As suggested, we have added the information of OGT-related XLID in the Introduction. It has been previously reported that OGT XLID mutants (e.g. L254F, A259T, R284P, A319T and E339G) form functional dimers with decreased thermal stability and exhibit impaired glycosyltransferase kinetics. We have analyzed the locations of these OGT missense mutations in the structure (Fig. 2a). These mutations affect residues in the TPR region that do not directly contact OGA. We speculate that they may distort the TPR helices and decrease their conformational elasticity, leading to defective substrate recruitment and modification. We have added discussion along this line in the revised manuscript.

5. *A more detailed comparison of their structure to the existing Cryo-EM and X-ray structures of OGT and OGA...especially focusing on any differences (there were some important differences with the X-ray structure and recent Cryo-EM structure of OGT--does your data support the recently observed Cryo-EM differences?)*

Response: We thank the reviewer for the suggestion. We have added a detailed comparison of our structure to the reported cryo-EM and X-ray structures of OGT and OGA (Supplementary Fig. 3c). The catalytic domain GTD₅₂₀₋₁₀₄₆ of our OGT dimer is almost identical (RMSD=1.096 for the backbone C-alpha atoms) to that of the reported OGT dimer (Meek et al. Nature Communications, 2021; PDB code: 7NTF), while the TPR domain of the two structures shows visible differences, consistent with the conformational flexibility of this domain. We have also compared our OGT dimer to the X-ray structures of OGT with UDP. The structure of the catalytic domain of our OGT dimer is very similar to those crystal structures. The backbone RMSDs of between our structure and the two crystal structures (PDB codes 3PE3 and 4GYW) are 1.337Å and 1.320Å, respectively. We have also discussed the major structural differences between OGA bound to OGT and the free OGA.

6. The final paragraph while important is speculation on a very interesting proposed model. It should be made clear...something along the lines of "our data supports a working model in which..."...also, what future work could further test this model?

Response: We thank the reviewer for this great suggestion. We have revised the final paragraph as suggested and discussed future work that can further test this model.

7. Ref. 31 is biorxiv for 2-3 years without being published in peer reviewed literature...perhaps a reference to the Zachara laboratory that has used Cre/Lox KO cell lines for sometime now would be more appropriate.

Response: As suggested, we have removed Ref. 31.

Reviewer #3 (Remarks to the Author):

In this manuscript the authors use cryo-EM to model the structure of OGT as well as OGT with a known substrate: OGA. The experiments described in this manuscript could potentially give insights into OGT structure, substrate recognition and catalysis. The authors also make claims that the OGT/OGA complex acts in vivo to sterically inhibit the other enzymes and is critical for maintaining O-GlcNAc homeostasis. However, the authors do not satisfactorily test or have data support these conclusions. The major flaws with the manuscript center around biochemical characterization of the complex formed and its existence in situ. Major issues with specific figures and further suggested experiments are outlined below.

Response: We thank the reviewer for the constructive suggestions. We have conducted additional experiments to biochemically characterize the OGT-OGA complex and to support its existence in human cells. The new results have strengthened our original conclusions and improved our manuscript.

1. Gels/Blots presented in this manuscript are cut-off and do not contain a ladder to assess size. Entire gels/blots should be provided (can be in extended figures) and ladders for gels/blots should be included in all figures. These are necessary for interpretation. Further all experiments should be performed multiple times, quantified and statistically analyzed.

Response: We have indicated the positions of the molecular weight ladders in the figures (as is traditionally done in most publications) and provided the uncropped gels/blots and other necessary original data in the Source Data file. All enzymatic experiments have been repeated for at least three times. We have added the quantification statistics in the revised figures.

2. Figure 1C. what is the antibody being used to detect OGT? Is it against OGT or the Halo tag? The author's interpreted the decrease in OGA to indicate that OGT acts to stabilize OGA protein, however, there is no evidence of this.

Response: In Figure 1c, the mouse anti-Flag antibody (Sigma-Aldrich, F1804) was used to detect the endogenous human OGT tagged with the Halo9-Flag dual tag at its C-terminus by CRISPR-Cas9. We agree with the reviewer that we do not have direct evidence to suggest that

OGT stabilizes OGA in human cells and have removed our speculation. We now simply state that depletion of OGT caused a decrease of OGA protein levels for unknown reasons.

3. *From figure 1C the authors conclude in lines 104-107 that “these findings suggest that large pools of OGT and OGA might form stable complexes in human cells and depend on each other for stability” However, the authors do not perform any experiment to test this conclusion. The author’s need to assess OGA transcription by qRT-PCR at the various time points to show that the effect is not due to coordinate transcriptional regulation. Further to prove that the loss of OGA is due to protein stability the author’s need to inhibit the proteasome to show that OGA levels remain. Additionally, the authors never show that OGT and OGA actually form a complex in vivo, the authors need to do co-IPs in cells not just in-vitro to show that a stable complex is formed. Assessment of in vivo binding of OGT and OGA should also be performed under varying nutrient conditions that alter the level of UDP-GlcNAc as the authors conclude that this complex forms to inhibit O-GlcNAcylation under conditions of high UDP-GlcNAc.*

Response: Multiple published reports have shown that OGT interacts with OGA and OGA is a substrate of OGT in human cells. As suggested, we have performed co-IP experiment between OGT and OGA in human HEK293 cells, and showed that OGT and OGA can indeed pull down each other (Supplementary Fig. 9a-b). More importantly, mutations of OGT residues that interact with OGA in our structure disrupted the binding, indicating that at least a pool of OGT is bound to OGA in human cells using the interface observed in our structure. We agree with the reviewer that it would be interesting to test whether the OGT-OGA complex is regulated by nutrient conditions. On the other hand, because the major focus of our current study is the biochemical and structural characterization of the OGT-OGA complex, we feel that assessment of the in vivo binding of OGT and OGA under varying nutrient conditions is beyond the scope of our current study.

4. *All OGT activity assays should be repeated using additional substrates as it is difficult to make claims on the overall activity of OGT in a cell based on glycosylation of 1 substrate in vitro. Other good substrates include CKII or Nucleoporin 62.*

Response: We thank the reviewer for this great suggestion. YTHDF1 is an m6A mRNA-binding protein and has been identified as an OGT substrate. We have been studying the potential functions of O-GlcNAcylation of YTHDF1 in a related study. We have used this substrate in glycosylation assays, in addition to TAB1. As shown in Figure 1f and Supplementary Figure 4c-d, human OGT WT exhibited robust activity towards both TAB1 and YTHDF1. As negative controls, the catalytically inactive mutant OGT K852M had no activity. The monomeric mutant OGT 4A had decreased activity towards both substrates, suggesting that the dimerization of OGT promotes optimal modification of substrates, but is not strictly required for the activity of OGT.

5. *Throughout the paper the authors use a glycosylated-TAB1 for experiments. However, this TAB1 has multiple bands, of which are lower molecular weight than TAB1 as can be seen in their blots. In figure 4F the authors state that the increase in lower bands when incubated with OGA was due to de-glycosylation, however, as can be seen in the other figures glycosylated-TAB1 (when incubating TAB1 with OGT) runs the same as TAB1. It is unclear what all of these*

bands are. The authors need to provide a better test of OGA activity; please see PMID: 28472822 (Kim, Eun Ju., Chembiochem. 2017) for a review on quantitative OGA activity assays.

Response: In our study, we co-expressed human full-length TAB1 and wild-type OGT in bacteria (*E. coli*) and purified TAB1 using the His₆-tag affinity purification approach. When co-expressed with OGT in bacteria, TAB1 was O-GlcNAcylated by OGT at multiple sites. The multiple bands of TAB1 represented the multiple modified forms of TAB1. We have purified a new batch of glycosylated TAB1 protein, repeated this experiment, and observed similar results (Fig. 5f). In addition, we also used the glycosylated YTHDF1 as the substrate for OGA and found that OGT inhibited the activity of OGA towards glycosylated YTHDF1 (Supplementary Fig. 10).

Finally, as suggested by the reviewer, we have used the quantitative OGA activity assay to measure the activity of OGA towards the artificial substrate p-nitrophenyl-beta-N-acetylglucosaminide (PNP-GlcNAc). As shown in Fig. 5g, compared to OGA WT alone, the OGT–OGA complex had much weaker O-GlcNAcase activity towards PNP-GlcNAc. These new data support our original conclusion that OGT inhibits the activity of OGA. We thank the reviewer for this great suggestion.

6. The authors state that OGA acts to sterically inhibit OGT, however, this is never tested using a competition assay.

Response: As suggested by the reviewer, we have performed the competition assay (Figure 4d). Addition of the catalytically dead OGA D175N mutant decreased the O-GlcNAcylation of TAB1 by OGT (Fig. 4d), indicating that, by acting as an OGT substrate, OGA can compete effectively with TAB1.

7. The authors state that after OGT glycosylates OGA it remains in a complex (lines 174-175), this would suggest that OGA needs to be glycosylated in order to form a stable complex, the authors should test whether the catalytic mutant of OGT forms a stable complex with OGA.

Response: Our cryo-EM structure of the OGT-OGA complex clearly shows that OGA is modified at the S405 site with O-GlcNAc. Because the protein sample was prepared with UDP-GlcNAc at normal buffer conditions, our structure likely captured the post-catalysis state, i.e. the enzyme-product (EP) complex, which was metastable under our experimental conditions. We do not believe that the complex requires the O-GlcNAc modification of OGA for formation. Consistent with this notion, we found that the catalytically inactive K852M mutant of OGT cofractionated with OGA on size-exclusion chromatography, suggesting that this OGT mutant could still form a complex with OGA.

8. Figure 2G and extended figure 8B gel lanes are inappropriately cut off. Additionally, as mentioned above pull downs should be done by overexpression in cells not in-vitro.

Response: We have provided the original gels/blots in the “Source data” file. As suggested, we have performed the co-IP experiment and found that OGT and OGA indeed pulled down each other when co-expressed in human HEK293 cells (Supplementary Figure 9a-b). The OGT TPR

mutants (N94A/N97A, N196A, N230A/N233A) had much weaker binding to OGA, consistent with the *in vitro* binding assays.

9. In extended figure 8C. Why is there no endogenous OGT/OGA and very little O-GlcNAcylation in the cells without HaloPROTAC?

Response: We used human HEK293 cells with the endogenous OGT tagged with the Halo tag for this assay. Addition of HaloPROTAC depleted OGT-Halo and reduced O-GlcNAcylation levels (compare lanes 1 and 2 in the original Supplementary Fig. 8c). We transiently transfected pCS2-Myc plasmids encoding different OGT constructs (WT and TPR mutants) in these cells prior to the HaloPROTAC treatment. These Myc-tagged OGT proteins were expressed at levels much higher than that of the endogenous OGT, resulting in much higher overall cellular O-GlcNAc levels. Importantly, the expression levels of Myc-OGT WT and mutants were similar. Thus, it is valid to compare the relative activities of these overexpressed proteins.

10. In extended figure 8d. What is being used for analysis? Entire lane? Specific band? What statistics are used?

Response: In this figure, the entire lanes were used for analysis of the cellular O-GlcNAc levels. The relative cellular O-GlcNAc levels were normalized to the expression levels of the Myc-OGT proteins. The p values were determined using One-way ANOVA ($p < 0.0001$, ****; $p < 0.001$, ***; $p < 0.01$, **; $n = 3$ independent experiments).

11. Extended figure 9 and figure 1F show inconsistent results for the ability of OGT4A to glycosylate. Please repeat all experiments multiple times and quantify/use appropriate statistics.

Response: We have optimized the assays and obtained consistent data (Figure 1f and Supplementary Figure 4c-d). The monomeric OGT 4A mutant had reduced activity towards the two substrates, TAB1 and YTHDF1. Each of our experiments has been repeated at least three times. The representative gels are shown, with the intensities of the O-GlcNAc bands are quantified as indicated.

12. Throughout the paper OGT4A consistently runs higher than WTOGT or OGTH508A. However, in figure 3D, lanes 6-8 the WT or OGT508A is running higher than WT/HOGT508A has been (as can be seen when compared to lanes 2-4) and similar to OGT4A. Was OGT4A accidentally used in these lanes? If so, this figure is not interpretable.

Response: It is not uncommon for certain protein mutants to migrate slightly differently from their wild-type counterpart on SDS-PAGE. We checked the sequencing results and confirmed that all mutants were correct. The correct proteins were used in these assays. The apparent minor differences in migration patterns could be due to unknown technical reasons. For example, OGT WT migrated differently in the absence or presence of OGA.

13. As mentioned above; the authors have no data to support the model in figure 5B or in their discussion.

Response: We have removed the speculative model depicted in previous Figure 5B and toned down our conclusions.

In summary, the findings have potential implications for understanding O-GlcNAc homeostasis but fall short in several significant ways. While some insight is gained from looking at the complex formed in vitro, the extrapolation to in vivo significance is problematic.

Response: We have added more data to support our original conclusions. We have focused on the discussion of the mutual regulation between OGT and OGA in vitro.

Reviewer #4 (Remarks to the Author):

This manuscript reports Cryo-EM structures of OGT and OGT/OGA complexes as well as biochemical studies that validate structural findings or explore possible implications of the structures. Given the importance of OGT, OGA, and O-GlcNAc cycling in mammalian cell biology (and metazoan biology in general), the structures reported are an important contribution to the field and justify publishing the manuscript.

Response: We thank the reviewer for his/her positive comments on our work.

However, before it can be published, the authors need to make revisions to the manuscript to address the following comments.

Major comments:

1. The authors draw a number of conclusions that are unwarranted based on the data presented. They should either carry out additional experiments to support currently unwarranted conclusions or modify the text. As appropriate, they should also acknowledge other interpretations of experimental results. See below for specific issues that need addressing.

a) The authors conclude that OGT stabilizes OGA in cells. This conclusion is based on two lines of evidence: 1) OGT forms a complex with OGA. 2) When OGT is degraded in cells, OGA becomes depleted. Because O-GlcNAc levels also decrease when OGT is degraded, and low O-GlcNAc levels are known to result in reduced OGA levels, the authors cannot conclude from their data that OGT directly stabilizes OGA. They will need to carry out additional experiments to draw this conclusion. Alternatively, they can speculate that OGT might stabilize OGA directly – but will need to acknowledge that there are other mechanisms by which OGT/O-GlcNAc regulate OGA expression. For example, inhibiting OGT leads to reduced OGA by mechanisms that include O-GlcNAc-dependent transcriptional control (transcription and splicing mechanisms - see 10.3389/fendo.2014.00206 and 10.1093/nar/gkaa263). OGT levels are also regulated by OGA via transcriptional mechanisms (that do not rely on OGT/OGA complex formation), and by splicing mechanisms that are dependent on O-GlcNAc levels (see 10.1074/jbc.RA118.004709 and 10.1016/j.celrep.2017.07.01, respectively). Reference 32 cited in the manuscript also identifies additional mechanisms for regulation of OGT and OGA levels. Given the flimsy support for a physical stabilization model, the authors should remove from the abstract assertions that OGT directly stabilizes OGA.

Response: We agree with the reviewer that OGT might regulate the levels of OGA through multiple mechanisms, including direct stabilization, post-translational modifications, indirect transcriptional and splicing controls, and other mechanisms. We have added data to demonstrate a physical interaction between OGT and OGA in human cells and toned down our conclusions in our revised manuscript.

b) The authors speculate that OGT and OGA form pools of complexes in cells that have a mutually regulatory effect. Consistent with this, cellular OGT can pull down OGA. However, OGT has many other interaction partners, and some pull down more reliably with OGT than OGA does (e.g., HCF-1, TRAK1). It is premature to propose that OGA negatively regulates OGT activity by binding to OGT when other proteins also bind to OGT, especially given that there is no quantitative information about how tightly OGA and various other proteins bind to OGT. The authors either need to do more experiments (including in cells) to support their model for OGA regulation of OGT activity, or they need to revise the manuscript to reduce emphasis on OGA's role in negative regulation of OGT.

Response: We agree with the reviewer that not all OGT molecules are bound to or regulated by OGA in human cells. Our structures and biochemical data in vitro clearly indicate that OGT when bound to OGA cannot modify other substrates. OGT and OGA can bind to each other in human cells. Thus, at least a pool of OGT forms a complex with OGA in cells. We have toned down our conclusions in the revised manuscript.

c) Regarding the model of mutual regulation through physical interaction: it is worth noting here that the authors' model would predict that OGA and OGT are colocalized in cells. They are commonly found in the same compartment, but how much of one colocalizes with the other has not been rigorously investigated (to my knowledge). However, there is one subcellular location where OGA is present and OGT is reported to be excluded: nucleoli. This observation undercuts the generality of their (speculative) model. I would urge the authors to exercise greater caution and not overinterpret the implications of their structures.

Response: Again, we agree with the reviewer that not all OGT molecules are bound to or regulated by OGA in human cells. We have toned down our conclusions in the revised manuscript.

d) Regarding the topological model in which the OGT TPR superhelix unwinds to accommodate entry of substrates: maybe unwinding occurs – that is one solution for how substrates with disordered regions sandwiched between folded domains can enter the lumen. However, there is no support for the unwinding model (as the authors acknowledge). The authors should soften the language asserting that a major unwinding of the TPR domain is required for substrate entry. In addition, the model appears to assume that substrates interact with the full length of the TPR lumen, but so far only a very small number of substrates have been investigated for their dependency on the TPR domain – and the results suggest that not all substrates bind to the full TPR domain.

Response: We agree with the reviewer that not all substrates bind to the full TPR domain. As reported by Joiner et al. (Joiner et al. Protein Substrates Engage the Lumen of O-GlcNAc Transferase's Tetratricopeptide Repeat Domain in Different Ways. *Biochemistry*. 2021. 60(11):847-853. doi: 10.1021/acs.biochem.0c00981), substrates engage the lumen of OGT's TPR domain in different ways. There are four possible types of substrates. The first type, like OGA, requires the distal TPR units to engage OGT. The second type of substrates needs the medium and proximal, but not distal TPR units for interactions with OGT whereas the third just engages the proximal TPR units. The fourth type of substrates might be recruited by OGT's catalytic domain but not the TPR domain. Our proposed model applies to OGA-like substrates that have long segments of intrinsically disordered regions (IDRs) and engage the entire TPR domain of OGT. We have modified the text accordingly in the revised manuscript.

e) The authors also propose that unwinding is required to release substrates, and they further propose that enclosing substrates in the TPR lumen allows processive glycosylation to occur. An interesting thing about OGT is that it often glycosylates only one or two ser/thr residues in long polypeptides in which many such residues are present. So for what substrates are the authors proposing a processive model? They should clarify this point because the text seems to suggest that OGT processively glycosylates a lot of substrates. (OGT was suggested to processively glycosylate the CTD of RNA Pol II, which is heavily glycosylated. However, an investigation showed that glycosylation of this substrate is distributive. See 10.1021/acs.biochem.5b01280.) If the authors are unable to identify multiple references that contain experiments demonstrating processive glycosylation of substrates by OGT, they should remove this line of speculation.

Response: As suggested by the reviewer, we have removed this speculation. The topological entrapment of OGA-like substrate is expected to prolong the enzyme-substrate interaction. This might help to explain why we could capture the post-catalysis state of the OGT–OGA complex. The functional consequences of this type of interaction are unclear at present.

2. I don't understand a key aspect of the model in Fig. 5b. Can the authors explain more clearly how high O-GlcNAc levels allow OGT substrates to outcompete OGA for binding to OGT? What might be the physical mechanisms that allow this to occur?

Response: Our structure captures the OGT–OGA complex in the post-catalysis state, with UDP bound at the catalytic site of OGT. We speculated that, at high concentrations of UDP-GlcNAc, UDP-GlcNAc can outcompete UDP for binding to OGT and displace the O-GlcNAcylated OGA from the complex. This frees OGT for binding and modifying other substrates. On the other hand, we agree with the reviewer that this is indeed simply a speculation. We have removed this model and revised the discussion in the revised manuscript.

Related to the model: the authors suggest that OGA negatively regulates OGT by binding to it stably enough to prevent glycosylation of 'weak' potential substrates; however, they propose that substrates that bind more tightly than OGA can displace OGA and be glycosylated. But if these good substrates bind tightly enough to OGT to displace something that binds tightly enough to pull down with OGT, wouldn't that present a problem for turnover? I might be missing something, but the suggested model doesn't completely make sense.

Related to testing the model: I understand the utility of picking a single substrate to use in experiments, but the authors should not make broad claims based on one substrate. The authors use TAB1 throughout the manuscript as a substrate for their in vitro biochemistry. But not all substrates are equal – see Joiner et al. (10.1021/acs.biochem.0c00981) and Shen et al. (10.1074/jbc.M111.310664). The authors need to tone down all claims that rely on using a single substrate as the 'test pool' for glycosylation activity.

Response: We agree with the reviewer that tight binding between OGT and OGA would potentially cause a turnover problem. It is possible that a slow turnover is indeed desirable for OGA-like substrates. Alternatively, there are yet unknown in vivo mechanisms to promote product release from OGT. We have discussed these possibilities in the revised manuscript. As suggested by the reviewer, we have added another substrate YTHDF1 in O-glycosylation assays and obtained similar conclusions.

3. Another problem with the manuscript is that the authors do not clearly put their findings in the context of other work. In several places, they seem to claim credit for findings that are very similar to findings of others, but without attribution. Some examples:

a) The authors make a dimer-disrupting mutant of OGT and report that the mutant is less active at glycosylating the substrate TAB1 than wildtype OGT. Previously, Davies and coworkers, who published the first cryoEM structure of OGT (Meek et al. ref 28 in the manuscript), reported that a dimer-disrupting OGT mutant was less active at glycosylating TAB1. Because the results here are the same as those in the Meek et al. paper, this TAB1 experiment should not be included in the manuscript – certainly not as a main text figure panel. The minor modifications in experimental design (e.g., these authors use a different dimer-disrupting mutant and assay glycosylation differently than did Davies and coworkers) do not justify including Figure 1f. The authors should move it to the SI, note that their results agree with previous studies, and cite the previous work in discussing effects of the dimer. In doing so, the authors should take care to be accurate. Currently, they state, “OGT functions as a homodimer” (lines 99-100); however, data in the Meek et al. paper show that substrates that extend far up into the lumen are sensitive to dimer status (TAB1), but shorter substrates (one based on an HCF-1pro repeat) are not.

Response: We agree with the reviewer that OGT dimerization might not be required for shorter substrates, such as peptides. We have cited the previous study and we have modified our statement to reflect this fact.

b) Asparagine ladder mutants were previously shown to have attenuated glycosylation activity in vitro and in cells. The authors should appropriately reference earlier examples of work reporting attenuated glycosylation activity for asparagine ladder mutants in vitro and in cells. Levine et al. reported in vitro studies showing that a five asparagine ladder mutant showed¹ (10.1073/pnas.2016778118). Other asparagine ladder mutants were also reported to have attenuated glycosylation activity using both purified protein substrates, including TAB1, and cell extracts (10.1021/acs.biochem.0c00981). And Kositzke et al. (ref. 29) studied interactions of OGA with individual luminal asparagine mutants, but this is not mentioned even though the reference is cited in a different context.

Response: We have now cited these previous studies as suggested.

c) *The authors should specifically comment on whether their OGA results agree or not with findings of Kositzke et al., who similarly reported that Asns in different positions have different effects on OGA binding (Kositzke et al. used crosslinking as a proxy for binding affinity, but the comparison should still be made).*

Response: We thank the reviewer for this great suggestion. Kositzke *et al.* applied the GEP1A fluorescence assay and identified 15 TPR residues that impacted OGT-OGA binding and/or sugar transfer. Our study is in general agreement with that of Kositzke *et al.* Both studies identified similar TPR hotspots for OGA binding and modification, especially for the asparagine ladder residues: N94 in TPR3, N196 in TPR6, and N230 in TPR7. We have added this citation in the text in the revised manuscript.

d) *The authors note that asparagine ladder residues contact the OGA backbone to anchor it in the TPR lumen. A crystal structure reported in 2013 showed that asparagine ladder residues can anchor peptides in the TPR lumen by making backbone contacts (10.1126/science.1243990). This was the first structure showing a peptide (HCF-1pro) bound in the TPR lumen and should be cited where the authors describe how OGA's disordered region binds to the TPR domain of OGT because it informed understanding of luminal binding. (And actually, it included the first in vitro experiments on Asn ladder mutants.) Are similar contacts made to the peptide backbone of HCF1pro as to OGA in the part of the lumen that both structures contain? (After correcting for the ten residue offset in the sequence numbering, it looks like several of the asparagine backbone contacts are similar.) A subsequent structural study fused part of the disordered region of TAB1 to OGT to obtain a structure showing contacts to the TPR lumen (10.1098/rsob.170078). A comment might be warranted if the contacts are consistent – although because this is a covalent fusion with a TPR-truncated OGT, the positioning may be biased. With the previous and the current structures showing different peptides bound in the TPR lumen, it is reasonable for the authors to make generalizations for how the asparagine ladder anchors peptides in the lumen. A comparison of OGT contacts to side chains of OGA and the HCFpro peptide used in the previous study might also allow some statements to be made about how selectivity is achieved. But the authors need to put their findings in context because other papers (cited above) have specifically described the role of asparagine ladder residues in backbone anchoring and have also identified other side chains as important in selectivity (via side chain contacts). It does not detract from the value of the structures reported in this manuscript to acknowledge earlier work that focused on how substrates bind in the TPR lumen and reached broadly similar conclusions. Indeed, appropriate comparisons would likely enrich their analysis.*

Response: Lazarus *et al.* (10.1126/science.1243990; HCF-1 Is Cleaved in the Active Site of O-GlcNAc Transferase) reported the first crystal structure showing a peptide (HCF-1pro) bound in the TPR lumen of OGT^{4,5}. Rafie *et al.* (10.1098/rsob.170078; Recognition of a glycosylation substrate by the O-GlcNAc transferase TPR repeats) reported the crystal structure of OGT^{4,5} with fused TAB1 peptide. The recognition of the HCF-1/TAB1 peptide by OGT mainly involved the conserved asparagine residues interacting with the amides of alternating residues along the peptide backbone, which is consistent with what we observed from the OGT-OGA structure. We thank the reviewer for this suggestion and have added these two citations in the text in the revised manuscript.

4. Rigor/technical issues/questions:

a) As noted above, the authors show that depletion of Halo-OGT results in depletion of OGA and conclude that OGT-OGA complexation stabilizes OGA. To evaluate whether OGT itself or O-GlcNAc regulates OGA abundance, the authors should consider providing a similar time course monitoring OGA levels when O-GlcNAc is lost through inhibition of OGT. A very good inhibitor of OGT exists (10.1021/jacs.8b07328) and can be purchased (OSMI-4). It would be useful to know the extent to which OGA levels decrease upon OGT active site inhibition, which results in decreased O-GlcNAc -which is known to affect OGA expression and functional mRNA abundance. (That said, a confounding factor could be that inhibitor binding in the active site displaces OGA, which would complicate interpretation of the results. The authors would need to look at this.) Alternatively, or in addition, OGA levels should be monitored when a catalytic dead variant of OGT is put into cells and Halo-OGT is degraded. Replicates and quantitation need to be performed. But if the authors really want to confirm their model, they should perform pulse-chain experiments that monitor OGA degradation rates under normal conditions and when OGT is depleted, and should provide some quantitation/estimate of OGT-OGA complex pools in cells. Otherwise, they should alter the text.

Response: We agree with the reviewer that not all OGT and OGA molecules form stable complexes in the cell. Direct stabilization is one among several possibilities that can explain the decreased OGA levels upon OGT depletion. We have toned down our conclusions in the revised manuscript.

b) Extended Data Fig. 1a: what is the gel below the gel filtration profiles supposed to show? None of the lanes are labeled and the caption does not describe it.

Response: The gel below the gel filtration profiles showed the fractions that contained the appropriate proteins or complexes. We have now labeled the fraction numbers on the gel.

c) Extended Data Fig. 1b: These data are described as showing that the H508A mutant does not glycosylate TAB1 at all. However, the amount of H508A used in the experiments appears to be ~25% the amount of WT OGT, which complicates interpretation. Moreover, there appears to be signal in the last lane of the WB (there is a disruption in the band that could be due to a transfer artifact). This experiment should be repeated using similar levels of WT and H508A mutant and replicates should be performed. (It is worth pointing out that the same mutant was tested in 10.1126/science.1243990 and shown to have some activity for glycosylating the CKII peptide. See SI – H498A mutant corresponds to the H508A mutant. The mutant does not appear to be fully catalytically dead. K852M, used later in the paper, is a more reliable catalytic dead mutant. Also, molecular weights should be indicated on the gels/blots.

Response: As suggested, we have made the OGT K852M mutant and used it as the catalytically dead control in the revised manuscript. The molecular weights are now indicated on gels and blots.

d) The western blot quality in Fig. 3d is poor and this experiment needs to be repeated. It is not possible to tell if there is faint signal in other lanes (than WT) because the background is so dark.

Response: We have adjusted the background of the blot and presented the new figure in the revised manuscript.

e) Fig. 4d: why are there such different amounts of OGT present in lanes containing OGT? Why does the G-TAB1 CBB signal look so different in lane 6 (from the left)?

Response: We used the same amount of OGT proteins in the assay. The G-TAB1 CBB signal looks different in lane 6 (from the left), because the active OGA has removed at least some of its O-GlcNAcylation and the hypo-GlcNAcylated TAB1 protein runs as smaller species on the gel. We have repeated the experiment and presented the new data in the revised manuscript.

f) Extended data Fig. 8c/d: how were the relative O-GlcNAc levels quantified? Did the authors correct for Myc-OGT levels? (And can OGT and Myc-OGT be distinguished by size?) More details on how quantitation was performed are required to understand the experiment. Also, the y-axis in the plot (8d) is misleading because the three identical divisions represent three different quantities (0.6, 0.2, and 0.4). The authors should make a proportionate linear scale from 0 and annotate at regular intervals. Data for the three independent replicates used to generate the plot should also be shown. The authors should clarify if independent replicates means different transfections.

Response: We have made the new figure to adjust the y-axis with a linear scale. The entire lane was used for analysis of the cellular O-GlcNAc levels. The relative cellular O-GlcNAc levels was normalized by the Myc-OGT protein expression levels. The p values are determined using One-way ANOVA ($p < 0.0001$, ****; $p < 0.001$, ***; $p < 0.01$, **; $n = 3$ independent experiments). Independent experiments mean different transfections. Data for all three replicates are now shown in the figure.

g) Extended data Fig. 9: How was the relative quantitation performed? (I assume the numbers below the WB represent relative quantitation results.) Are there replicates for this experiment? Replicates should be performed, and quantitation should include errors.

Response: We have repeated the experiment at least three times and quantified the relative intensities in the revised manuscript.

h) Re: extended data Fig. 9: there is a mobility difference between WT OGT and all the mutants, regardless of the location of mutations. Can the authors explain the apparent difference in size?

Response: We also noticed the slightly different gel mobilities of OGT mutant proteins. It is not uncommon for certain protein mutants to run differently from their wild-type counterparts.

i) Why did the authors make the 4A dimer-disrupting mutant instead of using the mutant others have reported in the literature (W198E/I201D)?

Response: We performed our mutagenesis studies before the Meek et al. study was published and used our own strategy to design the mutant. Both studies identified a similar dimer interface. We note that W198 and I201 mutated in previous studies correspond to W208 and I211 in our study. Alanine substitution is more conservative as it does not introduce destabilization energy. Importantly, both mutagenesis strategies produced consistent results.

j) Why do OGA levels increase out to 9 hours as OGT and O-GlcNAc are depleted? Is the blot in Fig. 1c representative? If this is reproducible, how do the results fit with the proposed model?

Response: The blot is representative. We do not fully understand why OGA levels increased transiently between 2-9 hours after OGT depletion. Regulation of OGA levels by OGT likely occurs at multiple levels, with stabilization being one possible mechanism. We have toned down our conclusions in the revised manuscript.

k) Can the authors provide confirmatory evidence that Halo tag knock-in hit only the OGT locus?

Response: We knocked in the Halo-Flag dual tags into the endogenous OGT gene locus in HEK293 cells using the CRISPR-Cas9 genome editing technology. The single clones of the cell were screened by PCR sequencing and Western blotting with the anti-OGT and anti-Flag antibodies for the integration of the Halo-Flag tag cassette into the endogenous OGT locus. We only observed a single band that corresponded to OGT-Halo-Flag in total cell lysates. DNA sequencing further confirmed the correct targeting. The targeting strategy and the Western blotting results are now presented in Supplementary Fig. 1a-b. Of course, we cannot rule out the possibility of off-target insertion of the Halo tag into genomic regions that do not result in its expression. Importantly, even if this does occur, it will not affect our conclusions.

l) The description of complex preparation does not mention UDP-GlcNAc. Did the authors include UDP-GlcNAc in setting up the complexes, which show UDP and glycopeptide? Please clarify.

Response: We incubated the OGT-OGA complex with UDP-GlcNAc for 30 min prior to the preparation of cryo-EM grids. We have added this information to the revised manuscript.

m) For synthetic human genes, were nucleotide sequences codon-optimized for bacterial expression? The authors should clarify which genes were cloned from a human cDNA library and which were synthesized. Are the encoded proteins otherwise identical to the human proteins? The authors should include full CBB gels and WBs showing purified proteins.

Response: In our study, human *OGT*, *OGA*, and *YTHDF1* were cloned from human cDNA libraries. Human *TAB1* gene was synthesized without codon optimization. We have provided the original uncropped gels in the “Source data” file.

n) The authors used structure 3PE3 in model building, but their structure contains a glycopeptide. There are deposited structures with glycopeptides and thioglycopeptides (e.g., I think the first were 4GYw, 4GZ3). Any reason these were not used?

Response: In our study, the X-ray structures of human OGT (PDB: 1W3B and 3PE3) or OGA (PDB: 5UN9) were used as the starting models and docked into the final Cryo-EM maps with UCSF Chimera. Because the binding of glycopeptides does not induce large conformational changes, all structures of OGT (with or without glycopeptides) can be used as starting models. There were no specific reasons for not using 4GYW and 4GZ3. Using 4GYW or 4GZ3 as the starting models would have produced the same results.

o) The authors cite Lazarus et al. Nature 2011 when talking about glycopeptides but the structures in this paper did not include UDP-sugars or glycopeptides bound. A subsequent Lazarus et al. paper included those structures ((2012) Nat Chem Biol 8: 966-968).

Response: We have now referenced the Lazarus et al. study in the revised manuscript.

n) The manuscript is poorly referenced. Citations to previous work on the asparagine ladder (see above) are lacking, but the problem is more pervasive. For example, in lines 225-227, the authors write, "Different subsets of TPRs are required for the recognition of different substrates..." but do not provide any citations. Another example of a sentence where a citation is required (line 229-30): "OGT prefers to bind to and act on a long segment (at least 50 residues) of intrinsically disordered regions in substrates." And line 64-65: "Biochemical and structural studies on human OGT and OGA have provided critical insights into the mechanistic [sic] underlying their enzymatic activities towards peptide substrates." There are other examples of sentences that need citations and the authors should go through the manuscript and provide them.

Response: We have now cited the appropriate references in the revised manuscript.

Other:

--There are a fair number of typos in the manuscript and the authors should go over it more carefully and correct errors in main text, methods, and extended data.

Response: We have gone through the manuscript and corrected the typos.

--The authors refer to the "asparagine ladder" within the OGT TPR lumen in talking about asparagines that face into the lumen. Canonically, the asparagine ladder comprises the conserved Asn residues at the 6th position within each TPR (some of the refs cited above describe the Asn ladder). The authors should clarify which backbone contacts are from conserved Asn ladder residues and which are from other Asn residues. A comment on the conservation of other residues that contact the OGA backbone could be useful.

Response: As suggested, we have now defined the OGA-contacting asparagines as the canonical asparagine ladder residues and other conserved asparagines. Both groups contact the backbone of OGA.

To sum up: The structure of the OGT-OGA complex justifies publishing this manuscript in a high-impact journal. There are novel aspects to the structure of the complex, including that it is the first structure showing a polypeptide bound throughout the full length of OGT's TPR lumen. The structure and accompanying biochemical experiments also imply that at least some of the time OGT and OGA are found together in inactive complexes, suggesting a regulatory role for complex formation - although the data supporting the proposed regulatory mechanism are inconclusive (and the model in 5b for how regulation is achieved is vague). In any case, substantial revisions are required to: 1) acknowledge/discuss previous work in the field that connects to the authors' findings; 2) tone down unsupported models/conclusions; 3) provide additional support for key conclusions; 4) address technical issues; 5) provide adequate citations.

Response: We thank the reviewer again for the extensive and constructive suggestions. As described above, we have addressed these specific concerns. As a result, the manuscript has been greatly improved.

REVIEWER COMMENTS

Reviewer #3 (Remarks to the Author):

Much improved manuscript given the recommendations of reviewers.

Reviewer #4 (Remarks to the Author):

As I noted in the previous review, this manuscript should be published for the OGT-OGA structure it reports, which is an important contribution to the field. However, further revision is necessary. Although the authors have addressed many of the reviewers' comments from the previous review and the manuscript has been improved over the original version, some claims are still not supported and the authors were not responsive to some of the comments. In addition, they have introduced some new issues in this revision that need to be addressed.

1. A major claim of the work, which is reflected in the title, is that OGT and OGA mutually regulate one another by forming a complex. Several reviewers questioned whether the data in the original paper supported this claim. The simple argument: If the majority of OGT in cells is not bound to OGA, OGA will not have much of an impact on OGT activity by binding to OGT; the same reasoning applies to OGT's effect on OGA activity in cells.

In response to the first reviewer's comment that the authors might be overselling their work by claiming mutual regulation, the authors state that "our conclusion about mutual regulation only applies to the stable OGT-OGA complex..." I recognize that the authors want to tie their structural findings to biology, but regulation is a concept that applies in cells and the authors have not provided any support for their claim. They have shown that the OGT-OGA complex is inactive in vitro towards glycosylation or deglycosylation of other proteins, a finding fully consistent with their structure. But these and other experiments presented in the manuscript do not allow them to conclude that OGT and OGA mutually regulate one another by forming complexes in cells. To do so, they need information about how much of the free proteins are present (or are bound to other proteins). It does not make scientific sense to simply state that their conclusion about mutual regulation "only applies to the stable OGT-OGA complex".

In response to the third reviewer's comment that they should confirm that OGT and OGA form a complex in cells, the authors showed that they can detect OGT and OGA via western blot in pulldowns

with OGA and OGT, respectively. These experiments do not support their mutual regulation argument because both constructs were tagged and transfected into cells, and no information was provided about the expression of the tagged constructs relative to the endogenous proteins. (It is unclear why the authors performed these experiments using overexpressed tagged constructs when antibodies exist for both OGT and OGA that have been used for pulldowns and western blots.) Moreover – and more important for their point - the authors did not comprehensively examine what other proteins are pulled down with each protein. Is OGA a major or minor component of the proteins pulled down with OGT under native or native-like expression conditions? In a recent quantitative study by the Zachara lab that identified many proteins known to interact with OGT, OGA was not found as a binding partner of OGT (10.1016/j.mcpro.2021.100069). Similarly, OGT was not found directly in an interactome study of OGA from the same lab. Other binding partners for these proteins are much more highly represented in unbiased pulldowns/interactome studies. The evidence in the literature overall suggests that the OGT-OGA complex is not particularly abundant compared to other complexes.

It remains possible that OGT and OGA form an inactive complex that is biologically relevant in a particular location in the cell, and that the authors can speculate about that. But they do not have evidence for the more global claim about mutual regulation – and evidence in the literature counters it.

In response to a third reviewer, who pointed out other pulldown data, the authors stated that they have “toned down our conclusions.” But revision to tone it down further is necessary because their claim is still not supported.

2. The authors should probably mention other OGT/OGA interactome findings in their discussion to put their work in context.

3. The authors did not respond to the request to move Fig. 1f to the SI. This request was based on the fact that another group has reported the same results – that an OGT mutant that cannot dimerize does not glycosylate TAB1 as well as WT OGT. The authors state that they cited the previous study, but this is not sufficient. If they want to keep that experiment in the main text, they need to explicitly state in the main text that another group has also reported that a dimerization-defective OGT mutant is less active than WT OGT at glycosylating TAB1.

(The authors also comment that they did not use the previously validated dimerization mutant because they performed their mutagenesis work before the Meek et al. study was published. But the dimerization mutant used in the Meek et al. study was originally published in 2004 when the TPR domain structure was published.)

4. In response to comments by several reviewers that Fig. 1c does not support their claim that OGT and OGA might stabilize one another in cells by interacting, the authors state the following in the revised manuscript: “HaloPROToc-mediated depletion of OGT resulted in the co-depletion of endogenous OGA in HEK293 cells, with a delayed kinetics (Fig. 1c). Similarly, depletion of OGA in human cells could also co-deplete OGT^{41,42}. These findings suggest that some pools of OGT and OGA might form complexes in human cells and depend on each other for stability^{20,39,43,44}.” This change is unresponsive. Halo-tagged OGT levels are substantially reduced by 2 hr whereas OGA levels increase at 2 hr and remain elevated over background for 9 hr. How do these findings “suggest that some pools of OGA and OGA...form complexes in human cells?” The manuscript would be stronger if the authors did not include this evidence as support for their claim because it doesn’t actually support their claim.

5. Western blots showing O-GlcNAcylation of substrate proteins in the presence of OGA need to show the region corresponding to OGA. Is it glycosylated as well? (one example: figure 4d)

6. The authors note that in their OGT-OGA structure, the difference between catalytic domains is ~ 10 Å less than in the OGT homodimer. Have the authors performed 3D flexibility or variability analysis on their cryoEM refinement structures to ensure that this is indeed the case?

7. The authors have included more description of how OGA binds in the TPR lumen along with some comparisons to other results in the literature, which is good. They also state that their results are largely consistent with other structures of substrates bound to the TPR lumen and to data on how OGA binds. However, in the main text and SI figures, several putative H-bonds greater than 3.5 Å are shown (e.g., Fig. 3a shows 4 putative H-bonds ranging from 3.9 to 4.8 Å; SI Fig. 7a shows putative H-bonds ranging from 3.8 to 5 Å – and there are multiple other examples in the manuscript). The authors cannot conclude that these are stabilizing interactions when the distances are longer than the normal cutoff for a H-bond.

8. In some places the authors show putative H-bonds that do not make chemical sense (even apart from the distances). For example, in SI Fig. 7b and 7c, the authors show a putative stabilizing contact from the carbonyl of N366 in OGT to the nitrogen of P416 in OGA. That nitrogen does not have a H on it and the indicated contact would presumably be repulsive. Similarly, one of the nitrogens of H29 of OGT is shown making putative contacts to two different carbonyls on OGA. Assuming that this His nitrogen, and not the other, bears the proton, it would be able to make only one H-bond to a carbonyl. The authors need to clean up their figures to remove these types of issues and to clarify what contacts they think are important.

Point-by-point response to reviewers' comments

Manuscript ID: NCOMMS-22-50787A

Previous Title: Mutual regulation of the O-GlcNAcylation enzyme pair revealed by Cryo-EM structure of human OGT–OGA complex

Current Title: Cryo-EM structure of human O-GlcNAcylation enzyme pair OGT–OGA complex

We thank the reviewers for their overall positive assessment of the importance of our study. Their constructive comments and suggestions are very helpful for the improvement of our manuscript. In response to their comments, we have further revised the text and figures. Our point-by-point responses to the reviewers are as followed. The reviewer's comments are in black and our responses are highlighted in blue.

Reviewer #3 (Remarks to the Author)

Reviewer #3 (Remarks to the Author):

Much improved manuscript given the recommendations of reviewers.

Response: We thank the reviewer for the positive comments on our revision.

Reviewer #4 (Remarks to the Author):

As I noted in the previous review, this manuscript should be published for the OGT-OGA structure it reports, which is an important contribution to the field. However, further revision is necessary. Although the authors have addressed many of the reviewers' comments from the previous review and the manuscript has been improved over the original version, some claims are still not supported and the authors were not responsive to some of the comments. In addition, they have introduced some new issues in this revision that need to be addressed.

1. A major claim of the work, which is reflected in the title, is that OGT and OGA mutually regulate one another by forming a complex. Several reviewers questioned whether the data in the original paper supported this claim. The simple argument: If the majority of OGT in cells is not bound to OGA, OGA will not have much of an impact on OGT activity by binding to OGT; the same reasoning applies to OGT's effect on OGA activity in cells.

In response to the first reviewer's comment that the authors might be overselling their work by claiming mutual regulation, the authors state that "our conclusion about mutual regulation only applies to the stable OGT-OGA complex..." I recognize that the authors want to tie their structural findings to biology, but regulation is a concept that applies in cells and the authors have not provided any support for their claim. They have shown that the OGT-OGA complex is inactive in vitro towards glycosylation or deglycosylation of other proteins, a finding fully consistent with their structure. But these and other experiments presented in the manuscript do not allow them to conclude that OGT and OGA mutually regulate one another by forming

complexes in cells. To do so, they need information about how much of the free proteins are present (or are bound to other proteins). It does not make scientific sense to simply state that their conclusion about mutual regulation “only applies to the stable OGT-OGA complex”.

Response: We thank the reviewer for the constructive comments. We do recognize that we should be more cautious when claiming the mutual regulation of OGT and OGA. Although our structural and biochemical findings on the OGT-OGA complex do suggest a mutual inhibition between OGT and OGA, it might not reflect the actual situation in the cells in vivo. With the helpful suggestion, we are happy to edit the title and text to tone down our claim on the mutual regulation of OGT and OGA. Firstly, we have changed the title of our manuscript to “Cryo-EM structure of human O-GlcNAcylation enzyme pair OGT–OGA complex”. Secondly, we have made corresponding changes to all the necessary parts in our revised manuscript.

In response to the third reviewer’s comment that they should confirm that OGT and OGA form a complex in cells, the authors showed that they can detect OGT and OGA via western blot in pulldowns with OGA and OGT, respectively. These experiments do not support their mutual regulation argument because both constructs were tagged and transfected into cells, and no information was provided about the expression of the tagged constructs relative to the endogenous proteins. (It is unclear why the authors performed these experiments using overexpressed tagged constructs when antibodies exist for both OGT and OGA that have been used for pulldowns and western blots.) Moreover – and more important for their point - the authors did not comprehensively examine what other proteins are pulled down with each protein. Is OGA a major or minor component of the proteins pulled down with OGT under native or native-like expression conditions? In a recent quantitative study by the Zachara lab that identified many proteins known to interact with OGT, OGA was not found as a binding partner of OGT (10.1016/j.mcpro.2021.100069). Similarly, OGT was not found directly in an interactome study of OGA from the same lab. Other binding partners for these proteins are much more highly represented in unbiased pulldowns/interactome studies. The evidence in the literature overall suggests that the OGT-OGA complex is not particularly abundant compared to other complexes.

It remains possible that OGT and OGA form an inactive complex that is biologically relevant in a particular location in the cell, and that the authors can speculate about that. But they do not have evidence for the more global claim about mutual regulation – and evidence in the literature counters it.

In response to a third reviewer, who pointed out other pull-down data, the authors stated that they have “toned down our conclusions.” But revision to tone it down further is necessary because their claim is still not supported.

Response: We thank the reviewer for the constructive comments. Our co-immunoprecipitation (co-IP) assays confirmed that OGT and OGA can pull down each other when co-expressed in human HEK293 cells (Supplementary Fig. 9a-b). For this assay, we used OGT with Myc tag and OGA with Flag tag, respectively. To clarify it more clearly, we have provided the experimental details in the “Methods” section and figure legends in our revised manuscript: “For in vivo Myc/Flag pull-down (Co-Immunoprecipitation) assay, 293FT cells were co-transfected

with pCS2-Myc/pCS2-Myc-OGT WT/mutant plasmids with pCS2-Flag/pCS2-Flag-OGA using Lipofectamine 2000 (Thermo Fisher Scientific) and cultured for 48 h in a 37°C incubator before collection by centrifuge. After cell lysis by sonication in TENT buffer (25 mM Tris pH 7.4, 150 mM NaCl, 0.5% TritonX100, 1 mM EDTA) with the addition of 1mM PMSF, protease and phosphatase Inhibitor, the cell lysate was centrifuged, and supernatants were separately incubated with anti-Myc and anti-Flag magnetic beads (LABELAD) for 2 h. After the removal of the unbound cell lysate by centrifuge, the beads were washed with TENT buffer and boiled with sample loading buffer for SDS-PAGE and western blot. The bait and bound prey were blotted with corresponding antibodies”. Since the major focus of our current study is the structural and biochemical characterization of OGT-OGA complex, we did not further examine what other proteins are pulled down with each protein. It is true that the Zachara lab did not identify OGA as a binding partner of OGT using mouse embryonic fibroblasts (MEFs) (10.1016/j.mcpro.2021.100069), which finds more than 130 proteins to interact with OGT. In terms of human OGT, over a thousand intracellular proteins (including OGA) are reported to be O-GlcNAcylated by OGT, and multiple studies indeed found out that human OGA is a substrate and in the interactome of OGT (10.1093/glycob/cwj078; 10.1093/glycob/cwj096; 10.1038/nchembio881; 10.1038/nchembio.412; 10.1016/j.ijbiomac.2020.12.078). The different results from different groups might be due to the different cell lines used and different detection approaches. We do not claim that OGT-OGA complex is in an abundant population in the cells and we have toned down our claim on the mutual regulation of OGT and OGA. Thus, we have changed the title of our manuscript to “Cryo-EM structure of human O-GlcNAcylation enzyme pair OGT–OGA complex”. In addition, we have made corresponding changes to all the necessary parts in our revised manuscript.

2. *The authors should probably mention other OGT/OGA interactome findings in their discussion to put their work in context.*

Response: We thank the reviewer for these constructive suggestions. We have added some sentences to mention the possibilities of OGT/OGA interactome in the Discussion part of the revised manuscript: “Although OGT and OGA do not always form a stable complex and have their own individual interactomes in the cells, human OGA is well recognized as a substrate of OGT *in vitro* and *in vivo*, suggesting that in certain situations OGT molecules are directly bound by OGA, and vice versa”.

3. *The authors did not respond to the request to move Fig. 1f to the SI. This request was based on the fact that another group has reported the same results – that an OGT mutant that cannot dimerize does not glycosylate TAB1 as well as WT OGT. The authors state that they cited the previous study, but this is not sufficient. If they want to keep that experiment in the main text, they need to explicitly state in the main text that another group has also reported that a dimerization-defective OGT mutant is less active than WT OGT at glycosylating TAB1.*

(The authors also comment that they did not use the previously validated dimerization mutant because they performed their mutagenesis work before the Meek et al. study was published. But the dimerization mutant used in the Meek et al. study was originally published in 2004 when the TPR domain structure was published.)

Response: We thank the reviewer for this constructive suggestion. We think the reviewer may mixed up the two publications we cited in the manuscript. Jinek *et al* (*Nature Structural and Molecular Biology*. 2004. 10.1038/nsmb833. *The superhelical TPR-repeat domain of O-linked GlcNAc transferase exhibits structural similarities to importin alpha*) is the one published in 2004, while Meek *et al* (*Nature Communications*. 2021. 10.1038/s41467-021-26796-6. *Cryo-EM structure provides insights into the dimer arrangement of the O-linked beta-N-acetylglucosamine transferase OGT*) was published at the end of 2021. We did our assays before the press of Meek *et al* study. We do appreciate that the dimerization-defective OGT mutants (W208AI211A) they made were well-prepared and can be used in enzymatic assays. Since all four residues (W208, L209, I211, and H212) are involved in OGT dimerization, we made OGT mutant with these four residues mutated. The OGT 4A (W208A/L209A/I211A/H212A) with four sites mutated, used in our study, should be able to reach similar, if not better effects in disrupting dimerization and enzymatic activities, as shown by our own biophysical and biochemical data. To better cite the published results, we have added the acknowledgment that both Jinek *et al* and Meek *et al* have shown that the dimerization-defective OGT mutants are less active towards substrates in the revised manuscript.

4. *In response to comments by several reviewers that Fig. 1c does not support their claim that OGT and OGA might stabilize one another in cells by interacting, the authors state the following in the revised manuscript: “HaloPROTOC-mediated depletion of OGT resulted in the co-depletion of endogenous OGA in HEK293 cells, with a delayed kinetics (Fig. 1c). Similarly, depletion of OGA in human cells could also co-deplete OGT41,42. These findings suggest that some pools of OGT and OGA might form complexes in human cells and depend on each other for stability20,39,43,44.” This change is unresponsive. Halo-tagged OGT levels are substantially reduced by 2 hr whereas OGA levels increase at 2 hr and remain elevated over background for 9 hr. How do these findings “suggest that some pools of OGA and OGA...form complexes in human cells?” The manuscript would be stronger if the authors did not include this evidence as support for their claim because it doesn’t actually support their claim.*

Response: We thank the reviewer for this constructive suggestion. In the revised manuscript, we have removed the OGA blot panel to better present Figure 1c, the major focus of which is to claim the essential role of OGT in cellular O-GlcNAcylation. In addition, in the revised manuscript we have removed the following sentences: “HaloPROTOC-mediated depletion of OGT resulted in the co-depletion of endogenous OGA in HEK293 cells, with delayed kinetics. Similarly, depletion of OGA in human cells could also co-deplete OGT. These findings suggest that some pools of OGT and OGA might form complexes in human cells and depend on each other for stability”. We have revised it as follows in the revised manuscript: “As a substrate of OGT, due to the limited structural information, how OGA is recognized and modified by OGT remains underexplored.”

5. *Western blots showing O-GlcNAcylation of substrate proteins in the presence of OGA need to show the region corresponding to OGA. Is it glycosylated as well? (one example: figure 4d)*

Response: We thank the reviewer for this suggestion. In the revised manuscript, we have shown the region of OGA in the revised Figure 3f, 4c, 4d, 5f, and Supplementary Figure 10. In fact, except for the OGA (371-440) region in binding assay, all the OGA protein used in the

enzymatic assay were in its full-length fragment. This assay (Figure 4d) was the competition assay to check the TAB1 O-GlcNAcylation by OGT with the presence of different doses of OGA. We focused on the TAB1 O-GlcNAcylation. Although we did not blot the O-GlcNAc level of OGA in this particular assay, we did have a similar repeating assay shown in the following figure. Indeed, OGA D175N is glycosylated in this assay. As shown by the data, the addition of the OGA (D175N) mutant decreased O-GlcNAcylation of TAB1 while increased its own O-GlcNAcylation by OGT, suggesting that OGA can act as a competitive inhibitor of OGT *in vitro*.

6. The authors note that in their OGT-OGA structure, the difference between catalytic domains is $\sim 10 \text{ \AA}$ less than in the OGT homodimer. Have the authors performed 3D flexibility or variability analysis on their cryoEM refinement structures to ensure that this is indeed the case?

Response: We thank the reviewer for this suggestion. Indeed, we have performed 3D classification for OGT-OGA complex. As shown by our 3D classification and refinement procedures in Supplementary Figure 5d, this analysis did not find classes with dramatic differences in the catalytic domains of OGT dimer. The most dramatic differences between different cryo-EM maps are the integrity of the volumes for OGA, but not the catalytic domains of OGT dimer. For all three cryo-EM maps for OGT-OGA complex, all the OGT molecules are bound by OGA, at least the flexible region (IDR) of OGA. Compared to the somewhat flexible OGA with weaker EM density, the OGT dimers within the complex are better modeled and relatively rigid. We also did 3D variability analysis for OGT-OGA complex and built the most different models among the generated 20 frames of EM maps. As shown in the following figure, the distance between the two catalytic domains is consistently smaller than that in the OGT homodimer (apo), suggesting that OGA binding induced the conformation changes between the two OGT protomers within the complex.

7. The authors have included more description of how OGA binds in the TPR lumen along with some comparisons to other results in the literature, which is good. They also state that their results are largely consistent with other structures of substrates bound to the TPR lumen and to data on how OGA binds. However, in the main text and SI figures, several putative H-bonds greater than 3.5 Å are shown (e.g., Fig. 3a shows 4 putative H-bonds ranging from 3.9 to 4.8 Å; SI Fig. 7a shows putative H-bonds ranging from 3.8 to 5 Å – and there are multiple other examples in the manuscript). The authors cannot conclude that these are stabilizing interactions when the distances are longer than the normal cutoff for a H-bond.

Response: We thank the reviewer for this constructive suggestion. Many types of interactions are involved between two protein molecules, including electrostatic, π -effects, van der Waals forces, and hydrophobic effects. A hydrogen bond (H-bond) is a specific type of electrostatic interaction, formed by the interaction of a hydrogen atom that is covalently bonded to an electronegative atom (donor) with another electronegative atom (acceptor). We agree with the reviewer that a common cutoff distance for energetically significant H-bonds in proteins is 2.7-3.3 Å (at most 3.5 - 4.0 Å for weak ones). The distance for van der Waals forces is usually 3.0 - 5.0 Å. Besides H-bonds interactions, other non-bonded/non-covalent interactions are also involved in the interfaces of OGT and OGA IDR. Non-bonded /non-covalent interactions are types of interactions that are not covalent and with rather long interatomic distances (2.0 - 5.0 Å) and energies usually much less than that of covalent bonds. We did not state that H-bond is the only type of interaction between OGT and OGA. Our data in Fig. 3a-c and Fig. S7a-c are to illustrate the most potential/possible (but not absolute) interaction pairs (including H-bonds and other non-bonded interactions) between OGT and OGA based on the solved structure. Based on the reviewer's suggestion, we have removed some distance labelings to avoid misunderstanding in the revised manuscript.

8. In some places the authors show putative H-bonds that do not make chemical sense (even apart from the distances). For example, in SI Fig. 7b and 7c, the authors show a putative stabilizing contact from the carbonyl of N366 in OGT to the nitrogen of P416 in OGA. That nitrogen does not have a H on it and the indicated contact would presumably be repulsive. Similarly, one of the nitrogens of H29 of OGT is shown making putative contacts to two different

carbonyls on OGA. Assuming that this His nitrogen, and not the other, bears the proton, it would be able to make only one H-bond to a carbonyl. The authors need to clean up their figures to remove these types of issues and to clarify what contacts they think are important.

Response: We thank the reviewer for this constructive suggestion. Based on the relative distance for the potential interactions, we have removed the dash lines between the carbonyl of N366 in OGT to the nitrogen of P416 in OGA (left), and between the nitrogen of OGT H29 and the carbonyl group of OGA S439 (right), in our revised manuscript.

REVIEWERS' COMMENTS

Reviewer #4 (Remarks to the Author):

The manuscript has been greatly improved through the revision process. The structure of an OGT-OGA complex is an important contribution.

(As an aside, this reviewer still thinks that showing putative hydrogen bonds greater than 3.5 Å is not appropriate. The authors state in their response letter that they did not call the depicted interactions hydrogen-bonding interactions – they note there are lots of different types of intermolecular interactions and did not mean to specify a particular type. However, most scientists would infer that a hydrogen bond is being represented by dashed lines drawn between an O and an N where one of these atoms bears a proton. On the other hand, the authors did remove the most egregious examples from their figures.

Overall, the revisions made this a much better paper and I congratulate the authors on a fine structure!)

Reviewer #5 (Remarks to the Author):

O-GlcNAc homeostasis is critical for cellular function. It has been proposed that mutual regulation between OGT and OGA is important for O-GlcNAc homeostasis. This study provides a crucial structural insight into the mutual regulation between OGT and OGA. The authors were responsive to the referee #4's remarks. The manuscript is acceptable for publication in the current form.